# Flex Linear Attention: Compiling a Unified Abstraction into Scalable Kernels for Linear Attention

**Haojie Duanmu**[*1 2 3], **Size Zheng**[†1], **Ningxin Zheng**[1], **Jianqiao Lu**[1], **Xuegui Zheng**[1]
**Xingcheng Zhang**[3], **Li-Wen Chang**[1], **Xin Liu**[1], **Dahua Lin**[3 4]

[1]ByteDance Seed  [2]Shanghai Jiao Tong University  [3]Shanghai AI Laboratory
[4]The Chinese University of Hong Kong

## Abstract

The quadratic complexity of softmax attention poses a major bottleneck for long-context modeling, motivating a surge of linear attention variants with linear complexity. Unlike softmax attention, which benefits from optimized kernels, linear attention lacks general-purpose, hardware-efficient support and scalable distributed implementations. We introduce **Flex**ible **L**inear **A**ttention (FlexLA), a domain-specific compiler that automates the generation of high-performance, scalable kernels for a wide range of linear attention models directly from high-level PyTorch code. At its core, FlexLA employs an intuitive programming abstraction that decomposes any linear attention algorithm into three canonical phases: intra-chunk computation, inter-chunk state propagation, and output merging. This unified abstraction enables FlexLA to perform domain-specific optimizations, automatically generating kernels that fuse computation and communication at a fine-grained tile level and eliminating host synchronization. Our evaluation demonstrates that FlexLA combines programmability with performance: a wide range of linear attention variants can be implemented in just a few dozen lines of code, while the generated kernels deliver 1.01x-4.9x the performance of sate-of-the-art expert-optimized library and scale with near-linear efficiency on scalar gated linear attention to 16 million tokens on 128 GPUs, surpassing the state-of-the-art distributed baseline by up to 7.2x.

## 1 Introduction

Transformer models rely on self-attention, which has quadratic time and memory complexity with respect to sequence length. As models handle increasingly long contexts, this quadratic bottleneck severely limits scalability. In response, many efficient-attention mechanisms have been proposed. In particular, linear attention methods remove the softmax nonlinearity and reorder computations to achieve linear computation complexity and constant-memory inference. This has led to a proliferation of innovative architectures, such as Mamba (Gu & Dao, 2023; Dao & Gu, 2024), RetNet (Sun et al., 2023), RWKV (Peng et al., 2023), GLA (Yang et al., 2023), HGRN (Qin et al., b) and Gated DeltaNet (Yang et al., 2024a). These architectures demonstrate capabilities competitive with, or even superior to, standard transformers.

Unlike softmax attention, which has benefited from highly optimized and now-standardized kernels like Flash-Attention (Dao et al., 2022) and Ring-Attention (Liu et al., 2023) that efficiently map its computation and communication to modern AI infrastructure, the landscape for linear attention is far more fragmented. Flash-Linear-Attention (FLA) (Yang & Zhang, 2024) provide a valuable collection of triton kernels for a series of linear attention variants. But it essentially relies on expert developers to provide manual implementation for each variants. The rapid evolution of linear attention variants means that a one-size-fits-all solution does not exist. This forces researchers into a costly and inefficient cycle of manual kernel development for each new variant, a process fraught with two major challenges.

---

[*]Work done at ByteDance Seed.  [†] Corresponding Author.

**PyTorch Implementation**

```python
def scalar_gla(q, k, v, g, s, o, scale):
    # q/k/v: [BHTD]. g: [BHT]. s: [BHDD]
    for bh in range(B * H):
        b, h = bh // H, bh % H
        state = s[b, h, :, :].clone()
        for chunk_idx in range(0, T, CHUNK_SIZE):
            c_q, c_k, c_v, c_g, c_o = get_chunk(…)
            # 1. compute intra-chunk decay
            g_cumsum, g_sum = c_g.cumsum(0), c_g.sum(0)
            decay_sum,decay_vec=g_sum.exp(), g_cumsum.exp()
            decay_k = (g_local_sum-g_cumsum).exp()[None, :]
            # 2. compute intra-chunk state
            c_k_decay  = (c_k * decay_k)
            curr_state = (c_k_decay @ c_v)
            # 3. compute output of current chunk
            decay_p = decay_vec[:, None]-decay_vec[None, :]
            p = ((chunk_q @ c_k.T).tril(0) * decay_p)
            curr_state_ = curr_state * decay_vec[:, None]
            c_o = (p @ c_v)+(c_q @ curr_state_) * scale
            # 4. update state
            state = state * decay_sum + curr_state
    return o
```

**Our DSL**

```python
def chunk_mode(k, v, g):
    g_cumsum, g_sum = g.cumsum(0), g.sum(0)
    g_cumsum       = (g_sum - g_cumsum[None, :]).exp()
    k_decay        = (k.T * g_cumsum.exp())
    return k_decay @ v

def decay_mode(old_s, s, g):
    return old_s * g.sum(0).exp() + s

def merge_mode(q, k, v, g, s, scale):
    decay_vec = g.cumsum(0).exp()
    decay_p   = decay_vec[:, None] - decay_vec[None, :]
    qk        = (q @ k.T).tril(0)
    p         = (qk * decay_p)
    return (p @ v + q @ s * decay_vec[:, None]) * scale

LinearAttention(
    SP_GROUP,
    chunk_mode,
    decay_mode,
    merge_mode,
)
```

**Performance**

Scalar GLA: 256K Sequence Length

Figure 1: Comparison between PyTorch and our proposed DSL in writing scalar gated linear attention (data type conversion is omitted for simplicity). And the performance comparison of different approaches. Test shape: 32 heads with 128 head dimension. Torch-Eager fails to parallelize the computation, while Torch-Compile also performs poorly. FLA provides expert optimized triton kernels, while program generated by FlexLA offers comparable performance to handwritten kernels with the scalability to distributed environments.

First, the implementation of high-performance kernels is an arduous task requiring deep hardware expertise. While many linear attention models share a conceptual similarity, their specific state update rules and memory access patterns can differ substantially. Achieving hardware efficiency necessitates not only fusing the state update rule into a single kernel but also manually tuning hardware-specific parameters like pipeline schedules and tile sizes. Even with high-level DSLs like Triton (Tillet & Cox, 2019), developers must often delve into low-level hardware details, such as managing barriers or dealing with shared memory capacity limitation, to extract maximum performance for each variant. This creates a high barrier to entry and slows down the pace of innovation.

Second, existing solutions lack robust support for distributed execution, which is non-negotiable for scaling to contexts of hundreds of thousands or millions of tokens. FLA offers a bunch of single-device kernels but do not address the distributed scaling problem. When sequence lengths exceed the memory capacity of a single accelerator, distributed sequence parallelism becomes essential. However, enabling sequence-parallel linear attention is non-trivial: it typically requires custom communication schedules tailored to each variant's state update rule. Existing sequence parallel schemes, such as LASP and LASP-2 (Sun et al., 2024a; 2025) are designed for specific architectures and employ generic communication primitives (e.g., `All-Gather` from NCCL). The mismatch of existing communication primitive and dataflow of distributed linear attention leading to significant network bandwidth underutilization(Chou et al., 2025).

**Can we provide a solution to bridge the gap between the rapid evolution of linear attention algorithms and the difficulty of developing scalable kernels?** We observe that many of these difficulties stem from not exploiting the common structure underlying linear-attention variants. Our central insight is that most linear-attention variants share a small set of canonical operations and data exchanges. Based on this, we introduce **Flex**ible **L**inear **A**ttention (FlexLA), a compiler-driven framework that allows implementing the majority of linear attention variants in a few lines of idiomatic PyTorch code and scaling them to distributed system. As shown in Figure 1, our DSL expresses linear attention in three modular functions. The compiler translates the DSL into high performance kernels: reducing latency from $34.6$ seconds (PyTorch eager) to $9.2$ ms, even better than SOTA hand-written kernel from FLA. More importantly, the latency was further reduced to $2.7$ ms when scaled to 4 GPUs distributed system.

The frontend of FlexLA ingests a user-defined computation logic for a linear attention variant. Its backend then intelligently maps this logic, along with potential communication operations, onto hardware accelerators and network interfaces, applying domain-specific optimizations to generate high-performance, distributed-aware kernels. FlexLA is built around three key principles: ❶ **A Linear-Attention-Specific Programming Abstraction.** We formulate our programming abstraction based on the canonical chunk-parallel representation of linear attention. This model decomposes the computation into three intuitive phases: a compute phase that processes local chunks of the sequence in parallel, an update phase that communicates and updates the inter-chunk states, and

a merge phase that combines the global state with local chunk results. This abstraction aligns directly with the mathematical structure of chunk-wise parallel form, allowing researchers to translate their algorithm's formulation into our framework with minimal effort by providing simple PyTorch callable. ❷ **Native Compute-Communication Fusion.** At compilation time, FlexLA lowers the three user-provided callables into Triton code. We leverage Triton-Distributed (Zheng et al., 2025a) as our compiler backend, which extends Triton with native communication primitives. This allows our compiler to generate fine-grained, tile-level communication instructions that are fused directly with computation. By creating custom communication patterns tailored to the algorithm, we bypass the overhead and limitations of standard libraries like NCCL, enabling a more efficient use of the underlying network fabric and dramatically improving hardware utilization. ❸ **Targeted Optimization of System Bottlenecks.** Beyond kernel fusion, we identify and optimize other critical system-level bottlenecks that affect real-world system performance. FlexLA employs a suite of techniques, including Ahead-of-Time (AOT) compilation with static kernel dispatcher built on top of Triton to reduce runtime overhead and an adaptive parallelism scheduler that dynamically explores the optimal configuration of compute resources, further boosting the end-to-end efficiency of linear attention execution.

To demonstrate its flexibility, we implement a broad range of linear attention variants using FlexLA, each requiring only dozens of lines of code. This programmability does not come at the cost of performance. On a single GPU, our generated kernels achieve 1.01x to 4.9x the performance of the state-of-the-art FLA library of expert-tuned kernels. Furthermore, in distributed settings, FlexLA demonstrates near-linear scalability on up to 128 GPUs, outperforming the leading open-source baseline by up to 7.2x.

## 2 PRELIMINARY

### 2.1 LINEAR ATTENTION ARCHITECTURE

Given a sequence $\mathbf{X} = [\mathbf{x}_1, \mathbf{x}_2, \ldots, \mathbf{x}_L]^\top \in \mathbb{R}^{L \times d}$, the input of attention block: $\mathbf{q}_i, \mathbf{k}_i, \mathbf{v}_i = \mathbf{W_q}\mathbf{x}_i, \mathbf{W_k}\mathbf{x}_i, \mathbf{W_v}\mathbf{x}_i$ where $\mathbf{x}_i, \mathbf{q}_i, \mathbf{k}_i, \mathbf{v}_i, \mathbf{y}_i \in \mathbb{R}^d$ and the weights $\mathbf{W_q}, \mathbf{W_k}, \mathbf{W_v} \in \mathbb{R}^{d \times d}$. Transformers employ softmax attention as a token mixer (Vaswani et al., 2017):

$$\mathbf{o}_i = \sum_{j=1}^{i} \frac{\exp(\mathbf{q}_i^\top \mathbf{k}_j)}{\sum_{p=1}^{i} \exp(\mathbf{q}_i^\top \mathbf{k}_p)} \mathbf{v}_j \qquad (1) \qquad\qquad \mathbf{O} = \mathrm{softmax}(\mathbf{Q}\mathbf{K}^\top \odot \mathbf{M})\mathbf{V} \qquad (2)$$

Equation 2 is the matrix form of Equation 1 where $\mathbf{Q} := [\mathbf{q}_1, \ldots, \mathbf{q}_L]^\top$, $\mathbf{K} := [\mathbf{k}_1, \ldots, \mathbf{k}_L]^\top$, $\mathbf{V} := [\mathbf{v}_1, \ldots, \mathbf{v}_L]^\top \in \mathbb{R}^{L \times d}$ and $\mathbf{M} \in \{-\infty, 1\}^{L \times L}$ is a causal mask.

Such matrix form is well suited to modern accelerators, which excel at large matrix multiplications, but it incurs $\mathcal{O}(L^2 d)$ complexity. If we remove the softmax operation, the computation becomes associative: $\mathbf{o}_i = \mathbf{q}_i(\mathbf{k}_i \mathbf{v}_i^\top)$ which reduces the complexity to $O(Ld^2)$. The recurrence form is expressed as:

$$\mathbf{S}_t = \mathbf{S}_{t-1} + \mathbf{k}_t \mathbf{v}_t^\top, \quad \mathbf{o}_t = \mathbf{q}_t \mathbf{S}_t. \qquad (3)$$

Here $\mathbf{S} \in \mathbb{R}^{d \times d}$ is the state (or memory) updated in each time step. Equation 3 highlights the key idea of linear attention: replacing the exponential kernel in softmax attention with a linear recurrence. Although this formulation resembles RNNs (Hochreiter & Schmidhuber, 1997). The critical difference is that dependencies across time steps remain *linear*, which makes parallel training possible. Indeed, linear attention can be written in fully parallel form:

$$\mathbf{O} = (\mathbf{Q}\mathbf{K}^\top \odot \mathbf{M})\mathbf{V} \qquad (4)$$

Modern linear attention often augments the recurrence with a decay or gating mechanism, e.g., $\mathbf{S}_t = \mathbf{G}_t \odot \mathbf{S}_{t-1} + \mathbf{k}_t \mathbf{v}_t^\top$ (Gu & Dao, 2023), or with more sophisticated update rules such as the delta rule (Yang et al., 2024b): $\mathbf{S}_t = \mathbf{S}_{t-1}(\mathbf{I} - \beta_t \mathbf{k}_t \mathbf{k}_t^\top) + \beta_t \mathbf{v}_t \mathbf{k}_t^\top$, which enhance memory utilization.

## 2.2 CHUNK-WISE PARALLEL FORM OF LINEAR ATTENTIONS

The fully parallel form in Equation 4 achieves maximum hardware utilization but retains quadratic complexity. Conversely, the recurrent form in Equation 3 has linear complexity but is inherently sequential and hardware-inefficient. In practice, linear attention strikes a balance by adopting chunk-wise parallelization (Hua et al., 2022; Sun et al., 2023; Yang et al., 2023).

Specifically, the sequence of length $L$ is partitioned into $\frac{L}{C}$ chunks of size $C$. Let $\mathbf{Q}_{[i]}, \mathbf{K}_{[i]}, \mathbf{V}_{[i]} \in \mathbb{R}^{C \times d}$ denote the query, key, and value matrices of the $i$-th chunk, and let $\mathbf{S}_{[i]} \in \mathbb{R}^{d \times d}$ be the state after processing chunk $i$. The chunk-wise formulation separates computation into intra-chunk and inter-chunk two parts, and then merge both together to get the final output as shown in Figure 2.

$$\mathbf{S}_{[i]} = \mathbf{S}_{[i-1]} + \underbrace{\sum_{j=(i-1)C}^{iC} \mathbf{k}_j^\top \mathbf{v}_j}_{\textbf{Matrix: } \mathbf{K}_{[i]}^\top \mathbf{V}_{[i]}} \qquad (5)$$

$$\mathbf{O}_{[i]} = \underbrace{\mathbf{Q}_{[i]} \mathbf{S}_{[i-1]}}_{\text{inter}} + \underbrace{\left(\mathbf{Q}_{[i]} \mathbf{K}_{[i]}^\top \odot \mathbf{M}\right) \mathbf{V}_{[i]}}_{\text{intra}} \qquad (6)$$

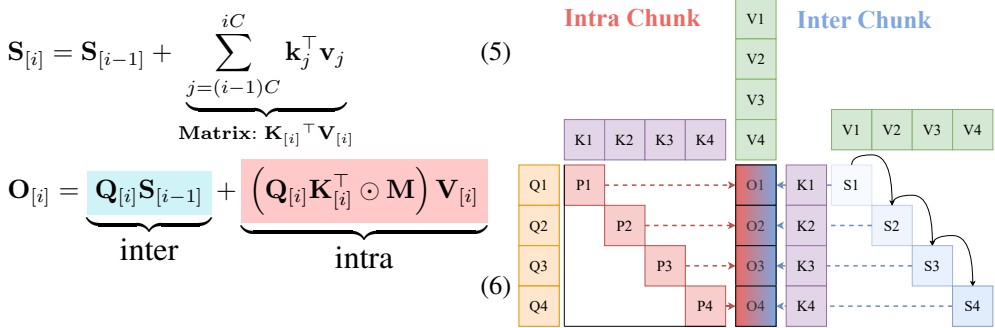

Figure 2: Chunk-wise parallel form demonstration for linear attention.

The inter chunk item can be viewed as readout memory from start of the sequence to the start of current chunk while the intra chunk item is processing the information in current chunk. When chunk size $C$ is set to sequence length $L$, it becomes the fully parallel form as in Equation 4. The chunk size is a tradeoff between parallelism and FLOPs.

## 2.3 DSLS AND DOMAIN SPECIFIC COMPILER

A domain-specific language (DSL) trades generality for performance by restricting program expressivity to a particular domain, creating opportunities for targeted optimization. Deep learning compilers like TVM (Chen et al., 2018), ThunderKitten (Spector et al., 2024), TileLang (Wang et al., 2025) and `torch.compile` (Ansel et al., 2024) exemplify this approach: by operating on a constrained set of primitives (i.e. tir in TVM and aten operators in PyTorch), they can systematically explore a focused design space to apply transformations like operator fusion and loop tiling, thereby automatically generating high-performance code from high-level descriptions.

This paradigm is particularly effective for operations that possess rich computational structure but high implementation complexity. By exposing domain-relevant abstractions, a DSL allows developers to specify *what* to compute, while delegating the complex details of *how* to execute it efficiently to the compiler. This separation of concerns is key to enabling automated, domain-specific optimizations without burdening the user with low-level hardware details.

Applied to the domain of linear attention, an effective DSL must therefore provide abstractions that are expressive enough to capture the diverse patterns found in various state update rules and parallel scan formulations. Simultaneously, its compiler must be able to recognize these common patterns and systematically generate optimized kernels for them, bridging the gap between high-level algorithmic design and performance, hardware-aware code. Because chunked parallel forms are complex to implement and offer opportunities to fully exploit hardware, this work focuses on the prefill phase of linear attention (used in inference or the forward pass of training). The backward pass can be implemented in a similar manner (Qin et al., a).

## 3 FLEXLA

In this section, we propose a unified abstraction of diverse linear attention variants. This abstraction enables programmers to easily express linear attention semantics without worrying about implementation details and kernel performance.

Table 1: A comparison of representative linear attention variants that can be easily mapped to our three-phase abstraction, including HGRN (Qin et al., 2023), RetNet (Sun et al., 2023), Mamba2 (Dao & Gu, 2024), GLA (Yang et al., 2023), and GDN (Yang et al., 2024a). Despite diverse state types (vector vs. matrix) and decay mechanisms (element-wise product vs. matrix multiplication). $v_t, k_t, q_t$ are value, key and query projections; $\alpha_t, \beta_t, r_t, i_t$ are gates; $\odot$ is the Hadamard product.

| Model | Update rule | Read-out | State + Decay type |
|---|---|---|---|
| HGRN | $h_t = \alpha_t \odot h_{t-1} + (1 - \alpha_t) \odot v_t$ | $o_t = h_t \odot q_t$ | vector + data-dependent vector |
| RetNet | $S_t = \gamma\, S_{t-1} + v_t k_t^\top$ | $o_t = S_t q_t$ | matrix + data-independent scalar |
| Mamba2 | $S_t = \gamma_t\, S_{t-1} + v_t k_t^\top$ | $o_t = S_t q_t$ | matrix + data-dependent scalar |
| GLA | $S_t = S_{t-1} \odot (1\alpha_t^\top) + v_t k_t^\top$ | $o_t = S_t q_t$ | matrix + data-dependent vector |
| GDN | $S_t = \alpha_t\, S_{t-1}(I - \beta_t k_t k_t^\top) + \beta_t v_t k_t^\top$ | $o_t = S_t q_t$ | matrix + data-dependent matrix |

## 3.1 Programming Abstraction

Despite the numerous linear attention variants designed by researchers, we unify these variants into three commonly shared phases based on chunk-wise parallel form introduced in subsection 2.2. ❶ Intra-Chunk Computation. The first phase computes a local state within each chunk of the input sequence. In this stage, computation across different chunks is embarrassingly parallel, as there are no data dependencies between them. Each chunk is processed independently, transforming its sequence of inputs into a relative state summary. ❷ Inter-Chunk State Propagation. The second phase addresses the dependencies between chunks. To compute the correct global state at the beginning of each chunk, the state summaries from all preceding chunks must be accumulated. For instance, in the case of vanilla linear attention, this propagation corresponds to a prefix sum (scan) operation, as shown in Equation 5. This phase is inherently sequential due to the temporal dependencies between chunk states. And cross-device communication happen in this phase. ❸ Merging and Output Generation. The final phase merges the results of the intra-chunk and inter-chunk computations. Here, operations are once again parallel across chunks. Each chunk utilizes the global state propagated from Phase 2 and its local inputs to compute its final output sequence.

Based on these insights, we designed our programming abstraction around three corresponding callable: `chunk_mode`, `decay_mode` and `merge_mode` correspond to three phases. This abstraction empowers users to implement a new linear attention variant by simply defining its chunk-wise parallel logic in idiomatic PyTorch code, decoupling algorithmic expression from system optimization. Furthermore, this three-phase decomposition also helps us optimize the program: we separate the parts of the entire program that can be executed in parallel and the parts that must be executed serially and may involve cross-device communication. With this information, FlexLA can perform more aggressive and accurate optimizations.

As shown in Table 1, prominent linear attention variants employ vastly different state representations and decay mechanisms at the token level. To map them to our abstraction, these token-level updates are reformulated into chunk-level matrix operations. For example, denoting the intra-chunk queries, keys, and values as matrices $Q, K$, and $V$, the chunk-wise parallel form for Mamba2 can be specified as:

$$S_{[t]} = \left(\prod \alpha\right) \odot S_{[t-1]} + V K^\top \qquad\qquad O = Q^\top K \odot M \odot G V^\top + S Q \qquad (7)$$

where $S_{[t]}$ is the state summary for the $t$-th chunk, $\alpha$ is the chunk decay, and $M$ and $G$ represent specific masking and gating matrices.

Under our framework, a user only needs to implement these equations within our three-phase programming abstraction in native PyTorch code. FlexLA then automatically handles all subsequent code generation, performance tuning, and scaling to distributed systems.

## 3.2 Compilation and Code Generation

Given the chunk-wise parallel form description in our DSL, FlexLA performs a series of graph-level and system-level transformations to produce optimized program as illustrated in Figure 3. The user-defined function is first captured as Torch.fx graph (Reed et al., 2022) via tracing. We choose fx graph as our intermediate representation (IR) because it is the most powerful tool in the PyTorch ecosystem, most torch operators can be directly captured, which makes our DSL

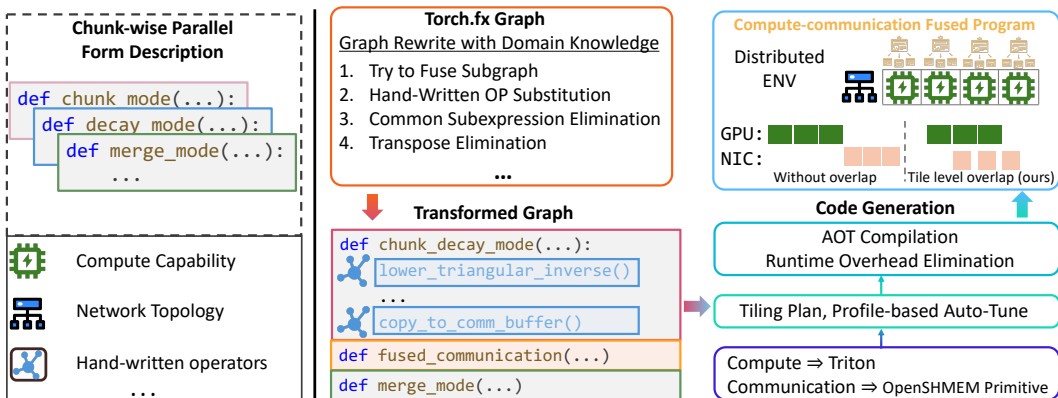

Figure 3: An overview of the FlexLA compilation pipeline. Our compiler ingests a high-level description of a linear attention algorithm, specified using our DSL (chunk_mode, decay_mode, merge_mode). This description is traced into a graph structure, which then undergoes a series of domain-specific optimization passes. The optimized graph is then compiled to Triton with native communication support, enabling fine-grained, tile-level overlap between computation and communication for better utilization of GPU and Network Interface Card (NIC).

expressive enough to implement most of linear attention variants. We first replace special op code in fx graph as custom instructions (e.g. placeholder is replaced with load instruction). Domain-specific optimization passes are then applied. To enable fusion of non-trivial operators (e.g. lower_triangular_inverse in GDN), we provide a bunch of custom triton kernels commonly used in linear attention domain and mapping them to corresponding torch operators. Once the compiler see these operators, for instance, torch.inverse, they will be marked and subsequently substituted with custom triton source code in code generation phase. Common optimization passes like transpose elimination are also applied in this phase. The IR is further rewritten with system-resource awareness. FlexLA take the hardware information to generate hardware-specific instructions. For instance, Tensor Memory Accelerator (TMA) availability is marked as an attribute of load instruction.

Recent studies have demonstrated that fine-grained compute–communication fusion can more effectively hide latency in distributed settings (Chang et al., 2024; Zheng et al., 2025b). We adopt this technique in FlexLA by automatically fusing computation and communication at the tile level, thereby reducing data dependency scope and eliminating the frequent GPU–host synchronizations inherent to traditional overlap strategies (Jangda et al., 2022). Within our programming abstraction, all cross-device communication is confined to the second phase, i.e., *inter-chunk state propagation*. Consequently, FlexLA first analyzes the data dependencies in this phase, then determines the corresponding computational tiling and communication tiling strategies based on the network topology, selects the appropriate communication mode, and generates computation and on-device communication instructions.

Finally, the IR is lowered into triton source code. Different with torch inductor (Ansel et al., 2024) targeting on official triton, FlexLA targets on Triton-Distributed (Zheng et al., 2025a), who addtionaly provide fine-grained communication control. On device computation is translated into computation primitives provided by Triton while the communication logic is mapped to the OpenShmem-style communication primitives provided exclusively by Triton-distributed, which are ultimately translated into GPU-initiated communication operations.

## 3.3 PERFORMANCE OPTIMIZATIONS

With domain-specific knowledge of linear attention, FlexLA can explore a compact yet effective optimization space. In particular, the choice of whether to fuse different phases introduces an important trade-off. For example, fusing chunk_mode with decay_mode avoids materializing intermediate states in global memory, thereby reducing memory traffic. However, such fusion also limits available parallelism, since computations can no longer be scheduled independently at the chunk level. There are many such trade-offs and FlexLA will handle all of these to get better performance. FlexLA employs a parallelism scheduling algorithm that dynamically chooses an optimal parallelization

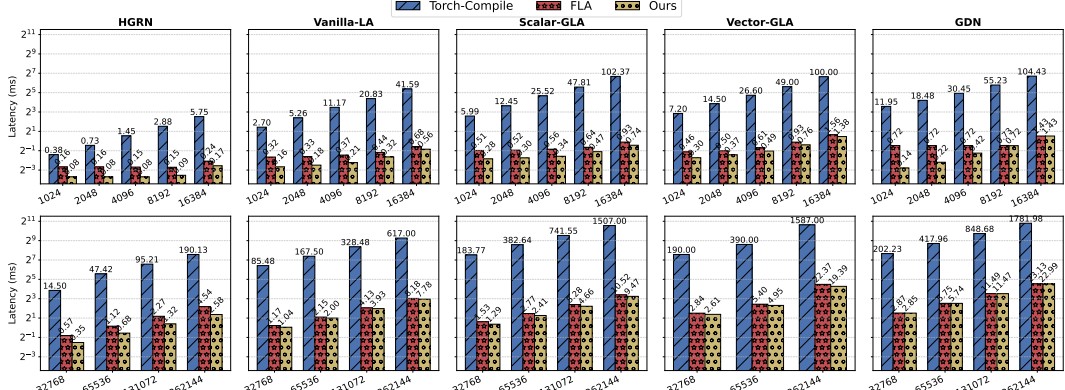

Figure 4: Latency comparison of different linear attention variants under varying sequence lengths on a single H100 GPU. Each subplot corresponds to one model, with the top row showing short to medium sequences (1K–16K) and the bottom row showing long sequences (32K–256K).

strategy based on input shapes and hardware information internally. The specifics of this parallelism scheduler are detailed in Appendix B.

Beyond the optimization for target program, we further incorporate a set of optimizations tailored to the compilation system itself. In practice, the Triton runtime introduces overheads on the order of hundreds of microseconds, often exceeding the actual most of linear attention kernel execution time at short to medium sequence length (e.g. 2K or even 4K). While approaches like CUDA-Graph can mitigate launch overhead for workloads with static input shapes, they often incur significant memory costs and are unsuitable for the dynamic workloads in inference scenarios. To address these limitations natively, we extend Triton compiler with a custom Ahead-of-Time (AOT) compilation module. Specifically, our module compiles Triton source code into pre-linked dynamic libraries ahead of execution. At runtime, FlexLA employs a profile-guided static dispatcher that bypasses the Triton runtime entirely, invoking the optimal pre-compiled binary directly through the CUDA Driver API. The static dispatcher is automatically generated by FlexLA from an offline performance database, ensuring that the empirically best-performing kernel is selected for any given workload without incurring runtime overhead from hash lookups or dynamic compilation logic.

Another problem is redundant compilation. Since our system is specialized for linear attention, we observe that certain input tensor dimensions (e.g., head dimension and number of heads) remain relatively static across runs, while sequence length is typically dynamic. This property facilitates efficient AOT compilation: FlexLA allows users to specify constant dimensions and their admissible ranges via input metadata. FlexLA enumerates the Cartesian product of these ranges and generates all potentially required kernels in advance. Furthermore, by leveraging PyTorch's symbolic tracing, our system supports tensors with symbolic shapes, ensuring that recompilation is unnecessary unless static dim change.

## 4 EXPERIMENTS

We implement several high-performance linear attention kernels using FlexLA, including HGRN, vanilla linear attention, scalar GLA, vector GLA, and Gated DeltaNet (Qin et al., 2023; Dao & Gu, 2024; Yang et al., 2023; 2024a). While many linear attention models differ in their parameterization, the computational patterns we implement cover over ten existing model designs (Appendix C).

### 4.1 SINGLE-DEVICE EVALUATION

Figure 4 reports the latency of kernels generated by FlexLA across sequence lengths ranging from 1K to 256K on a single H100 GPU. We compare with two baselines: Torch-Compile, representing a general-purpose compiler without domain-specific knowledge, and FLA (commit hash: 02766e71), the state-of-the-art library of providing expert-tuned Triton kernels for linear attention. For all variants except HGRN, we fix $\text{BatchSize} = 1$, $\text{NumHeads} = 32$, and $\text{HeadDim} = 128$, and vary sequence length. For HGRN, we follow its original single-head configuration. A batch size of

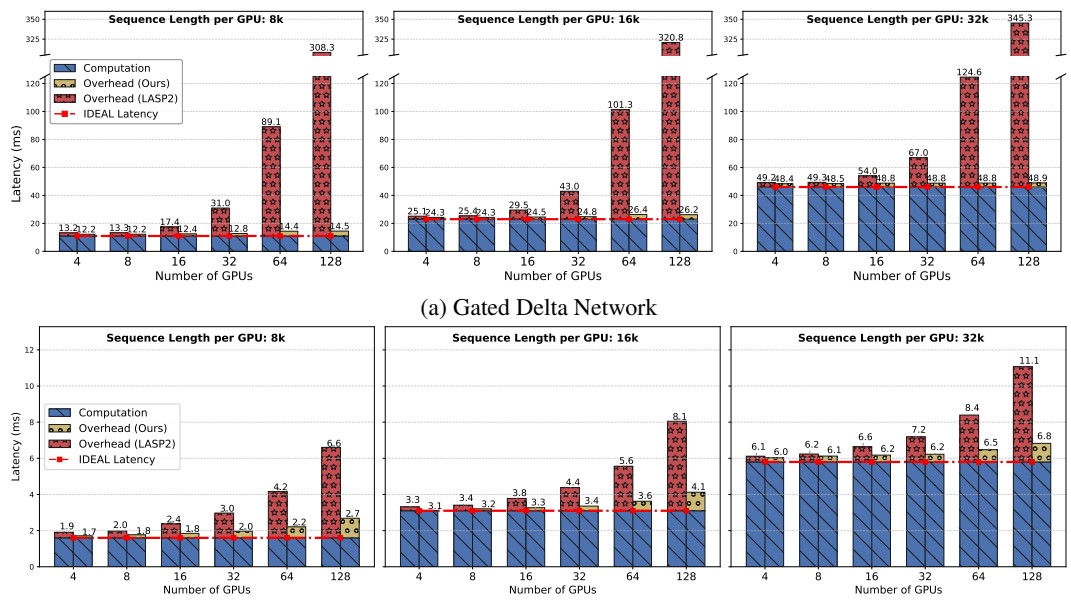

Figure 5: Weak scaling performance comparison for GDN and GLA models. Both figures show latency breakdown across a varying number of GPUs with a fixed sequence length per GPU.

one is a standard and reasonable choice, as a single long sequence in linear attention is computationally equivalent to a batch of packed shorter sequences.

Torch-Compile consistently exhibits poor performance, as it fails to apply the domain-specific fusion strategies required for linear attention. FLA achieves strong performance by carefully hand-tuning IO-aware tiling. Our work, FlexLA, automating this optimization process, consistently matches or outperforms FLA. On HGRN, FlexLA achieves 1.64–2.02× speedup, highlighting its ability to discover optimization opportunities beyond expert tuning (We check the generated code and find FlexLA use a more efficient threads allocation). On scalar and vector GLA as well as vanilla linear attention, FlexLA provides stable improvements (1.1–2.0×) while for GDN, performance converges to FLA at longer sequences. The speedup is most pronounced on short to medium sequence length. In this regime, system-level overheads and the choice of parallelism strategy are the dominant factors, and FlexLA effectively eliminate these bottlenecks, as discussed in subsection 3.3.

## 4.2 SEQUENCE PARALLEL EVALUATION

We further evaluate weak scaling behavior under distributed training, comparing against LASP2 (Sun et al., 2025), the strongest open-source baseline at the time of writing. ZeCO (Chou et al., 2025) reports improved scaling via pipelined communication, but its implementation is not public. We benchmark two representative workloads: GDN and scalar GLA, which feature matrix-multiplication and element-wise decay mechanisms, respectively. Experiments were conducted on a cluster of up to 128 NVIDIA H20 GPUs, interconnected with NVSwitch within nodes and Infini-Band between nodes. We fix $\text{BatchSize} = 4$, $\text{NumHeads} = 32$, $\text{HeadDim} = 128$, and maintain a constant workload per GPU while scaling the total sequence length from 128K (on 4 GPUs) to 4 million tokens (on 128 GPUs). The ideal outcome for weak scaling is a constant execution time.

Figure 5 show that FlexLA exhibits near-ideal weak scaling for both workloads: latency remains flat as GPU count increases. This is attributable to two core features of our compiler. First, it generates communication patterns that avoid the data redundancy incurred by the All-Gather primitive used in LASP2. Second, its ability to fuse computation and communication effectively hides the latency of the local state update. This is particularly impactful for GDN, whose matrix-based update is time consuming. In contrast, the communication and computation redundancy of LASP2 is amplified as the number of nodes increases, causing its performance to degrade significantly (e.g., from 49.2ms on 4 GPUs to 345ms on 128 GPUs for GDN). This confirms that FlexLA eliminates redundant communication and achieves scalable performance on large GPU clusters.

### 4.3 ABLATION STUDY

**AOT compilation with static dispatcher.** We measure the end-to-end latency of the execution of Scalar GLA kernel including both kernel execution and to demonstrate Triton runtime overhead and the efficiency of our solution as shown in Figure 6. At a sequence length of 1024, this overhead (207μs) is over 4.4 times the actual kernel execution time (47μs). Our static dispatcher mitigates this issue by reducing the overhead by 46%, yielding a 1.6x end-to-end speedup on these latency-sensitive inputs. As the sequence length grows and execution time becomes the dominant factor, our dispatcher consistently maintains a negligible overhead that is effectively eliminated at 8192 tokens (reducing from 101μs to just 1μs).

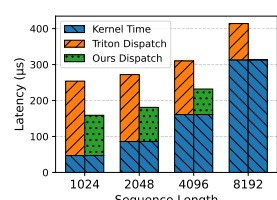

Figure 6: Execution time decomposition.

**Tile Level Compute-Communication Overlapping.** We conduct a targeted study to isolate and quantify the benefits of our tile-level compute-communication fusion. We set two baselines for the inter-rank state propagation: ❶ Serial communication, where each rank $i$ waits to receive all state data from rank $i - 1$ before performing its local update and sending to rank $i + 1$. ❷ Pipelined baseline, which chunks the state and uses standard NCCL send/recv operations to overlap the computation of one chunk with the communication of the next. Our method, in contrast, generates a single kernel that fuses the state update computation with communication primitives at the tile level. The results, presented in Table 2 show the standard Pipelined (PyTorch) baseline is slightly slower than the naive Serial implementation, demonstrating that host-managed pipelining can be counterproductive due to the significant overhead of launching numerous small operations and the required host-device synchronization. Conversely, our fused kernel substantially reduces the total time, achieving a 1.56x speedup over the serial baseline.

Table 2: State communication time on 8xH800 GPUs. State size is 67MB.

| Method | Time (us) |
| --- | --- |
| Serial | 873 |
| Torch-Pipeline | 902 |
| Ours | 560 |

## 5 RELATED WORK

There are a wide range of approaches to address the quadratic complexity of softmax attention. Sparse attention mechanisms (Zaheer et al., 2020; Xiao et al., 2023; Yuan et al., 2025) leverage structured or un-structured sparsity in attention to skip computation. Quantized attention (Shah et al., 2024; Zhang et al., 2024) use exploit low-precision arithmetic unit in modern hardware to get higher throughput. There are also various techniques to reduce key–value (KV) cache overhead. KIVI (Liu et al., 2024b) and SKVQ (Duanmu et al., 2024) directly compress KV cache using quantization while grouped-query attention (GQA) (Ainslie et al., 2023), and multi-head latent attention (MLA) (Liu et al., 2024a) alter the attention architecture to reduce memory overhead.

Another line of work proposes architectural alternatives with lower complexity, including linear attention variants (Katharopoulos et al., 2020; Dao & Gu, 2024; Peng et al., 2023; Yang et al., 2024a) as well as test-time-training approaches (Sun et al., 2024b; Behrouz et al., 2024). For linear attention sequence parallelism, LASP (Sun et al., 2024a) first extend linear attention to distributed environments with serial send-receive primitive. LASP2 (Sun et al., 2025) improves on this by leveraging collective communication primitives, yet both approaches still incur significant bandwidth under-utilization. ZeCO (Chou et al., 2025) introduces a pipelined send–receive scheme to hide send/receive latency but relies on manual chunk-size tuning and does not detail its implementation. Therefore, it was not included in our comparison.

AI compilers have been developed to optimize a broad range of workloads. `torch.compile` (Ansel et al., 2024), TVM (Chen et al., 2018), and TASO (Jia et al., 2019) are effective for common operators but their optimization spaces do not cover linear attention. Operator-level compilers such as Triton (Tillet & Cox, 2019), ThunderKitten Spector et al. (2024), TileLang (Wang et al., 2025), and Triton-Distributed (Zheng et al., 2025a) provide expressive abstractions for modern accelerators, but supporting the large and growing family of linear attention variants still requires substantial manual development effort. The most closely related

work, FlexAttention (Dong et al., 2024), targets block-sparse softmax attention and is limited to single-device settings.

## 6 CONCLUSION

We presented FlexLA, a domain-specific compiler for linear attention that unifies diverse algorithmic variants under a common three-phase abstraction. By generating hardware-efficient kernels with native distributed execution support, FlexLA bridges the gap between rapidly evolving linear attention research and the complexity of hand-tuned implementations. Our evaluation demonstrates that FlexLA achieves both high performance and broad applicability across modern linear attention models. We hope this work will accelerate the development of new architectures and inspire further research at the intersection of deep learning algorithms and domain-specific compilation.

## ACKNOWLEDGEMENTS

The authors would like to thank the diligent anonymous reviewers for their constructive feedback. We also extend our gratitude to the fla-org open-source community for their invaluable contributions to the linear attention infrastructure. This work is supported by the Shanghai Municipal Science and Technology Major Project.

## REPRODUCIBILITY STATEMENT

We are committed to the reproducibility of our work. Most of algorithms, implementation details, and experimental setups are described in the paper. Due to organizational policies requiring an internal review prior to public release, the source code is not included with the submission. However, we are committed to open-sourcing the code and will provide it to reviewers upon request for the purpose of evaluation.

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

## A  USAGE OF LLMS

For the preparation of this manuscript, we utilized LLM, as writing assistant to enhance the quality of the prose. Our process was interactive: the authors provided initial drafts and specific sentences to the LLM and used its suggestions to refine the text. Majority of prompts aimed at refining sentences to improve conciseness, in addition to correcting grammar and improving overall clarity. The core scientific ideas, methodology, and experimental results were developed exclusively by the human authors, who bear full responsibility for all claims and content within this paper.

## B  PARALLISIM SCHEDULER

FlexLA automatically explore different parallel schemes for linear attention during compilation. With the domain knowledge of linear attention, we can shrink the search space into two mainly used schemes, which is partially explored in FLA (Yang & Zhang, 2024).

In this section, we provide a theoretical analysis of two parallel execution strategies for linear attention using vanilla linear attention on GPU as an example. Then we derive a heuristic scheduling algorithm 1 that FlexLA used to build a parallelism scheduler, selecting the optimal strategy based on the input tensor shapes and hardware characteristics.

### B.1  NOTATION

The vanilla linear attention is defined by the recurrence:

$$\mathbf{S}_t = \mathbf{S}_{t-1} + \mathbf{k}_t \mathbf{v}_t^T \tag{8}$$

$$\mathbf{o}_t = \mathbf{q}_t \mathbf{S}_t \tag{9}$$

where $\mathbf{k}_t, \mathbf{q}_t \in \mathbb{R}^{D_k}$ and $\mathbf{v}_t \in \mathbb{R}^{D_v}$ are the key, query, and value vectors at timestep $t$, and $\mathbf{S}_t \in \mathbb{R}^{D_k \times D_v}$ is the state matrix. We consider batched inputs $\mathbf{Q}, \mathbf{K} \in \mathbb{R}^{B \times T \times H \times D_k}$ and $\mathbf{V} \in \mathbb{R}^{B \times T \times H \times D_v}$. The sequence of length $T$ is divided into $N_C$ chunks of size $C$, such that $T = N_C \times C$. Let $N_{SM}$ be the number of Streaming Multiprocessors (SMs) on the target GPU, representing its parallel execution capacity.

### B.2  STRATEGY 1: DECOUPLED THREE-PHASE EXECUTION

This strategy directly maps the chunk-wise parallel algorithm onto our three-phase programming model. Each phase is a separate kernel launch.

**Parallelism Analysis.**  The degree of parallelism varies significantly between phases.

- **Phase 1 (Intra-Chunk State Computation):** Computations for each chunk are independent. The total number of parallel tasks is $B \times H \times N_C$. This phase exhibits the highest degree of parallelism.

- **Phase 2 (Inter-Chunk State Propagation):** The prefix sum (scan) is sequential across the $N_C$ dimension. Parallelism is limited to the batch ($B$) and head ($H$) dimensions. Further parallelism, which we denote as $P_{state}$, can be achieved by partitioning the state matrix $\mathbf{S} \in \mathbb{R}^{D_k \times D_v}$ and processing its partitions on different thread blocks. The total number of parallel tasks is $B \times H \times P_{state}$.

- **Phase 3 (Merge):** Similar to Phase 1, the merge operation is independent across chunks, offering a high degree of parallelism with $B \times H \times N_C$ tasks.

**Memory Access Analysis.**  The defining characteristic of this strategy is the materialization of intermediate states in global memory (GMEM) between phases.

- **Phase 1:** Reads $\mathbf{K}$ and $\mathbf{V}$ from GMEM. Writes the intermediate, chunk-local states $\mathbf{S}' \in \mathbb{R}^{B \times H \times N_C \times D_k \times D_v}$ back to GMEM.

$$\text{GMEM Traffic}_{P1} = \underbrace{BHT(D_k + D_v)}_{\text{Reads}} + \underbrace{BHN_C D_k D_v}_{\text{Writes}} \tag{10}$$

- **Phase 2:** Reads the $B \times H \times N_C$ chunk-local states from GMEM, performs the scan, and writes the updated global states $\mathbf{S}_{global} \in \mathbb{R}^{B \times H \times N_C \times D_k \times D_v}$ back to GMEM.

$$\text{GMEM Traffic}_{P2} = \underbrace{BHN_C D_k D_v}_{\text{Reads}} + \underbrace{BHN_C D_k D_v}_{\text{Writes}} = 2BHN_C D_k D_v \qquad (11)$$

The total GMEM traffic introduced between Phase 1 and Phase 2 is $3 \times B \times H \times N_C \times D_k D_v$.

### B.3 Strategy 2: Fused Intra- and Inter-Chunk Execution

This strategy fuses Phase 1 and Phase 2, computing the local state of a chunk and immediately uses it to update the global state, all within on-chip memory.

**Parallelism Analysis.** By fusing the phases, the execution is constrained by the most sequential part, which is the inter-chunk scan. Therefore, the maximum number of parallel tasks is identical to that of Phase 2 in the decoupled strategy: $B \times H \times P_{state}$. The high parallelism across the chunk dimension ($N_C$) is sacrificed.

**Memory Access Analysis.** The primary advantage of this strategy is the elimination of the intermediate GMEM traffic.

- **Fused Phase (1+2):** Each of the $B \times H \times P_{state}$ parallel tasks loads its corresponding slice of $\mathbf{K}$ and $\mathbf{V}$ from GMEM chunk by chunk. The intermediate, chunk-local states are generated and accumulated entirely within SMEM. The only state written to GMEM is the final updated global state for each chunk, which is required by the Merge phase.

$$\text{GMEM Traffic}_{P1+P2} = \underbrace{BHT(D_k + D_v)}_{\text{Reads}} + \underbrace{BHN_C D_k D_v}_{\text{Writes}} \qquad (12)$$

Compared to Strategy 1, this approach saves $2 \times B \times H \times N_C \times D_k D_v$ worth of GMEM traffic, which is the cost of one full read and one full write of the intermediate state tensor.

### B.4 Heuristic Scheduling Algorithm

The choice between the Decoupled and Fused strategies presents a classic trade-off between parallelism and memory locality.

- **Strategy 1 (Decoupled)** is favored when the GPU has a high degree of parallelism ($N_{SM}$ is large) that is not saturated by the task parallelism of Strategy 2 ($B \times H \times P_{state}$). The performance gain from launching more parallel tasks in Phase 1 and 3 must outweigh the latency incurred by the extra GMEM I/O.
- **Strategy 2 (Fused)** is favored when the task parallelism of the scan ($B \times H \times P_{state}$) is already sufficient to saturate the GPU's SMs, or when the cost of reading/writing the intermediate states is the dominant performance bottleneck.

We can formulate a simple heuristic to guide this choice as shown in Algo 1.

In practice, we set $T_{\text{comp\_chunk}}$ to 0 to reduce the runtime overhead and omit the micro-benchmark effort. The insight is that most linear attention variants is memory bound instead of compute-intensive.

## C Computational Patterns and Model Coverage

This appendix elaborates on the claim that our implemented kernels for a few representative models can support a much broader range of linear attention variants. The mapping from our representative implementations to the computational patterns and the models they cover is detailed below and summarized in Table 3.

---

**Algorithm 1** Parallelism Scheduler)

---

**Require:** Shapes $B, T, H, d_k, d_v$, chunk size $C$, tile sizes $t_k, t_v$
**Require:** Device params: memory bandwidth $BW$ (bytes/s), $P_{\max}$ (hardware concurrency cap)
**Require:** Cost param: $T_{\text{comp\_chunk}}$ (measured per-chunk compute time in seconds)
**Require:** Thresholds: $P_{\min}$ (saturation threshold, tasks), element size $s$ (bytes)
**Ensure:** Return strategy $\in \{\text{FUSED}, \text{DECOUPLED}\}$
  1: $N_c \leftarrow \lceil T/C \rceil$
  2: $P_{\text{state}} \leftarrow \lceil d_k/t_k \rceil \cdot \lceil d_v/t_v \rceil$
  3: $P_{\text{dec}} \leftarrow B \cdot H \cdot N_c \cdot P_{\text{state}}$
  4: $P_{\text{fused}} \leftarrow B \cdot H \cdot P_{\text{state}}$
  5: $\text{GMEM}_{\text{dec}} \leftarrow \big(2C(d_k + d_v) + 5d_k d_v\big) \cdot s$
  6: $\text{GMEM}_{\text{fused}} \leftarrow \big(2C(d_k + d_v) + 4d_k d_v\big) \cdot s$
      $\triangleright$ Fast path: if fused already provides enough parallelism, prefer fused to save GMEM IO
  7: **if** $P_{\text{fused}} \geq P_{\min}$ **then**
  8:     **return FUSED**
  9: **end if**

                              $\triangleright$ Otherwise estimate per-chunk runtime under each mapping
 10: $\tilde{P}_{\text{dec}} \leftarrow \min(P_{\text{dec}}, P_{\max})$
 11: $\tilde{P}_{\text{fused}} \leftarrow \min(P_{\text{fused}}, P_{\max})$
 12: $\hat{T}_{\text{dec}} \leftarrow \dfrac{\text{GMEM}_{\text{dec}}}{BW \cdot \tilde{P}_{\text{dec}}} + \dfrac{T_{\text{comp\_chunk}}}{\tilde{P}_{\text{dec}}}$
 13: $\hat{T}_{\text{fused}} \leftarrow \dfrac{\text{GMEM}_{\text{fused}}}{BW \cdot \tilde{P}_{\text{fused}}} + \dfrac{T_{\text{comp\_chunk}}}{\tilde{P}_{\text{fused}}}$
 14: **if** $\hat{T}_{\text{fused}} \leq \hat{T}_{\text{dec}}$ **then**
 15:     **return FUSED**
 16: **else**
 17:     **return DECOUPLED**
 18: **end if**

---

**Vector-State Linear RNNs (represented by HGRN)**   Our HGRN kernel embodies the computational pattern of linear recurrent networks where the state is a vector, updated via element-wise operations with gated inputs. This is a common pattern for models aiming for high efficiency with a compact state. Models sharing this fundamental structure include the original HGRN (Qin et al., 2023) and Hawk (RG-LRU) (De et al., 2024).

**Matrix-State with Scalar Decay (represented by scalar-GLA)**   The scalar-GLA kernel represents the widely adopted matrix-state linear attention pattern. In this formulation, the state is a matrix updated via an outer product of key and value vectors, combined with a simple data-dependent or data-independent scalar decay. This pattern is foundational to many prominent and powerful models such as RetNet (Sun et al., 2023), Mamba-2 (Dao & Gu, 2024), and Lightning Attention (Qin et al., 2024).

**Matrix-State with Vector Decay (represented by vector-GLA)**   Our vector-GLA implementation captures the pattern of matrix-state models that employ a more expressive, data-dependent vector decay. This allows for per-feature state transition dynamics, offering a richer representation than a single scalar decay. Models in this category include the original Gated Linear Attention (GLA) (Yang et al., 2023), HGRN-2 (Qin et al., b), and RWKV-6 (Peng et al., 2023).

**Delta-Rule Updates (represented by GDN)**   Finally, our Gated DeltaNet (GDN) kernel is representative of a family of models based on the delta rule for state updates. This update mechanism can be interpreted as applying a series of Householder transformations to the state matrix, enabling more complex state transitions. This pattern is central to the family of DeltaNets, including the original DeltaNet (Yang et al., 2024b), GatedDeltaNet (Yang et al., 2024a), and DeltaProduct (Siems et al., 2025).

Table 3: Mapping of representative kernels implemented in subsection 4.1 to the broader set of models they cover. This demonstrates the generality of our compiler.

| Representative Kernel | Covered Models (Examples) |
| --- | --- |
| HGRN | HGRN (Qin et al., 2023), Hawk (RG-LRU) (De et al., 2024) |
| Scalar-GLA | RetNet (Sun et al., 2023), Mamba-2 (Dao & Gu, 2024), Lightning Attention (Qin et al., 2024) |
| Vector-GLA | GLA (Yang et al., 2023), HGRN-2 (Qin et al., b), RWKV-6 (Peng et al., 2023) |
| GDN | DeltaNet (Yang et al., 2024b), GatedDeltaNet (Yang et al., 2024a), DeltaProduct (Siems et al., 2025) |

## D    DISCUSSION ON EXTENSIBILITY AND SUSTAINABILITY

The longevity and utility of a Domain-Specific Language (DSL) depend heavily on whether its abstraction captures the invariant properties of the target domain rather than transient heuristics. In this section, we discuss the future-proofing of FlexLA from two perspectives: algorithmic expressiveness and hardware sustainability.

### D.1    ALGORITHMIC EXPRESSIVENESS: GROUNDED IN ASSOCIATIVITY

The universality of FlexLA's three-phase abstraction (Intra-Chunk, Inter-Chunk, and Merge) derives from the mathematical foundation of efficient sequence modeling: the **associative property**.

**Mathematical Invariance.**    Virtually all Linear Attention and State Space Duality (SSD) models aim to achieve $O(N)$ computational complexity by formulating the attention mechanism as a recurrence or a parallel prefix scan. Mathematically, any algorithm that can be decomposed into a chunk parallel form fits the FlexLA abstraction. This is not an ad-hoc design choice but a direct mapping of the underlying algebraic structure required for parallelization.

**Handling Complex Dependencies.**    FlexLA is expressive enough to handle complex state updates found in modern architectures, provided they satisfy chunk-wise associativity. For instance:

- **The Delta Rule:** Despite involving data-dependent updates (e.g., $h_t = h_{t-1} + \beta_t(v_t - h_{t-1})$), the Delta Rule preserves the associative structure within local chunks, allowing FlexLA to effectively fuse operations.
- **Element-wise Decay:** Varying decay rates (as seen in Vector-Gated GLA) are fully supported, as the element-wise multiplication distributes over addition, maintaining the scan property.

**Theoretical Limitations and Edge Cases.**    The boundary of FlexLA's applicability is strictly defined by **non-associative recurrences**. If an architecture introduces a dependency where the state update $h_t = f(h_{t-1}, x_t)$ involves a non-linear function $f$ that prevents parallel(e.g., passing the hidden state through a complex MLP at every step before the next update), it cannot be expressed in FlexLA.

A notable example is the Test-Time Training (TTT) layer (Sun et al., 2024b). While TTT can be viewed through the lens of linear attention or fast weights, its gradient-based updates involve non-linearities that break associativity. Consequently, TTT cannot be accelerated via the parallel prefix scans used in FlexLA. However, it is worth noting that this limitation is mutual: by abandoning associativity, such models fundamentally forego the massively parallel efficiency that characterizes the linear attention regime targeted by FlexLA.

### D.2    HARDWARE SUSTAINABILITY: HIERARCHICAL DECOUPLING

To ensure sustainability amidst rapid hardware evolution, FlexLA employs a strict hierarchical decoupling between the algorithmic description and hardware-specific instructions.

**Layered Compilation.** FlexLA operates as a high-level graph compiler rather than a low-level assembler. It does not directly emit hardware-specific assembly (ISA). Instead, it lowers the PyTorch-based algorithmic description into an intermediate kernel DSL. In our current implementation, we target `Triton`, effectively delegating low-level complexities such as register allocation, instruction scheduling, and tensor core management (e.g., `WGMMA` on Hopper) to the Triton compiler.

**Backend Agnosticism.** This layered design makes FlexLA inherently adaptable. While the current backend is Triton, the architecture allows for retargeting the mapping layer to other emerging DSLs (e.g., ThunderKitten (Spector et al., 2024) or TileLang (Wang et al., 2025)). These DSLs provide better performance with the cost of hardware overfit (e.g. ThunderKittens target on NVidia GPU, HipKittens (Hu et al., 2025) target solely on AMD GPU while Triton support multiple hardware backend). Triton allows us to minimize the use of inline assembly (e.g. PTX on NVGPU), reserving it only for specific extensions not yet exposed by the intermediate representation.

**Forward Compatibility.** Consequently, as hardware architectures evolve (e.g., the transition to NVIDIA Blackwell), FlexLA benefits automatically from updates made by the community to the intermediate compiler stack. This ensures that user-level code remains stable and performant without requiring modification, solving the maintenance bottleneck often associated with hand-written kernels.

## E    DISCUSSION ON GENERALIZABILITY TO SOFTMAX ATTENTION

While Forge is explicitly designed for Linear Attention, the underlying mathematical associativity, which enables our optimization is shared by Softmax attention (exploited via the online softmax trick (Dao et al., 2022)). In this section, we analyze the theoretical feasibility of extending Forge to support Softmax attention, and the design rationale behind our decision to specialize in the Linear Attention domain.

Extending the Forge abstraction to support Softmax-based mechanisms is theoretically feasible but would necessitate two primary modifications to the current three-phase abstraction:

- **Generalizing the Inter-Chunk State.** In Linear Attention, the state propagated between chunks is typically a feature map (e.g., $S \in \mathbb{R}^{d \times d}$) governed by a linear recurrence. In contrast, Softmax attention requires the propagation of normalization statistics, specifically the running maximum $m$ and the running sum $l$ to stabilize the computation of exponentials. Extending FlexLA would require modifying the `decay_mode` phase to support the propagation of these scalar or vector statistics alongside, or instead of, the matrix state.

- **Introducing a Rescaling Primitive.** The `merge_mode` in FlexLA is currently designed for linear combinations (accumulation). Softmax attention, however, mandates a renormalization step when merging partial results. Specifically, the output of a preceding block $O_1$ must be scaled down based on the difference between its local maximum $m_1$ and the new global maximum $m_{new}$:

$$O_{new} = O_1 \times e^{m_1 - m_{new}} + O_2 \times e^{m_2 - m_{new}} \tag{13}$$

Supporting this would require introducing a native `rescale(output, old_stats, new_stats)` primitive into the FlexLA DSL.I

**Despite the feasibility of these extensions, we deliberately narrowed the scope of FlexLA to Linear Attention to maximize expressiveness**.

**The Tension between Universality and Specificity.** Constructing a "universal" DSL often necessitates a rigid abstraction that compromises the ability to model complex, domain-specific patterns. A recent framework, AttentionEngine (Chen et al., 2025), attempts to unify both Softmax and Linear Attention. However, to maintain this broad generality, its abstraction struggles to express advanced Linear Attention variants that feature intricate dependencies, such as the **Delta Rule** or **Vector-Gated GLA**. By specializing in Linear Attention, Forge avoids these constraints, supporting these complex patterns effortlessly.

From an ecosystem perspective, Softmax attention is already well-served by highly optimized libraries and compilers (e.g., FlexAttention (Dong et al., 2024)). In contrast, the Linear Attention landscape is characterized by rapid algorithmic fragmentation, where new variants are proposed frequently but lack efficient kernel support. FlexLA addresses this specific "N-to-1" compiler challenge, solving a critical bottleneck that is currently more pressing for the research community than further optimizing Softmax kernels.

Finally, while the abstraction of FlexLA is specialized, the *system-level optimizations* are broadly applicable. For example, our Ahead-of-Time (AOT) compilation pipeline and static dispatcher, designed to eliminate runtime Python overheads, are universal optimizations. These techniques can be directly adopted by Softmax-focused DSLs (such as FlexAttention) to significantly improve performance, particularly in short-sequence regimes where dispatch latency is a dominant factor.

## F   EXPERIMENTS: DISTRIBUTED EVALUATION ON H100 CLUSTER

To verify the robustness of our distributed scaling strategy, we conducted additional experiments on an **8x H100 node** connected via NVLink.

Table 4 presents the latency comparison between Ring-Attention (a standard distributed softmax attention baseline) (Liu et al., 2023) with FlashAttention-3 which exploit advanced hardware feature on Hopper GPU, LASP-2, and FlexLA (both LASP2 and FlexLA using the Scalar GLA variant). The experiments were conducted with a **batch size of 1**, $\text{NumHeads} = 32$, and $\text{HeadDim} = 128$. The **SeqLen** in table means sequence length per GPU(global sequence length is equal to **SeqLen** $\times$ **NumberOfGPUs**).

The result shows FlexLA consistently outperforms the LASP-2 baseline across all sequence lengths. For instance, at a sequence length of 16k on 4 GPUs, FlexLA achieves a **1.17$\times$ speedup** over LASP-2 (0.96ms vs. 1.13ms). Even with the high bandwidth of NVLink on H100s, FlexLA maintains its efficiency advantage. This confirms that our fine-grained compute-communication overlap strategy is effective not only on bandwidth-constrained hardware (like the H20 used in the main result) but also on high-performance flagship clusters. As expected, Linear Attention methods (both LASP-2 and FlexLA) are orders of magnitude faster than Ring-Attention, especially at longer sequences (e.g., at 512k global sequence length, FlexLA is over **160$\times$ faster** than Ring-Attention).

These results validate that the performance gains reported in the main paper are not artifacts of the H20 hardware or larger batch sizes, but rather stem from the fundamental efficiency of Forge's generated kernels and scheduling logic.

Table 4: **Distributed Latency Comparison on H100 GPUs (Batch Size = 1).** We compare the end-to-end latency of Forge (Scalar GLA) against Ring-Attention (Softmax) and LASP-2 (State-of-the-art Linear Attention Sequence Parallelism). Forge consistently achieves the lowest latency across all sequence lengths and GPU configurations.

| # GPUs | SeqLen Per GPU | Latency (ms) | | |
| --- | --- | --- | --- | --- |
| | | Ring-Attn | LASP-2 | Forge (Ours) |
| 4 | 8192 | 8.40 | 1.02 | **0.79** |
| | 16384 | 17.77 | 1.13 | **0.96** |
| | 32768 | 61.65 | 1.66 | **1.57** |
| | 65536 | 229.43 | 2.90 | **2.76** |
| 8 | 8192 | 19.32 | 1.00 | **0.77** |
| | 16384 | 38.09 | 1.15 | **0.97** |
| | 32768 | 133.14 | 1.67 | **1.62** |
| | 65536 | 462.30 | 2.92 | **2.84** |

## G   CODE EXAMPLE

In this section we provide code examples to implement different linear attention variants and the code generate by FlexLA. Firt we show the Scalar GLA in Listing 1 and the generated code in List-

ing 2. FlexLA enable pre-compiled kernel (AOT) and static dispatch at runtime. In this example, we tune the kernel on H=[1,4,8,16,32] and D=[64, 128], so there are a lot of auto-generated code encoded pre-tuned information for dispatcher (our AOT optimization). Using FlexLA to implement this kernel only involves 50+ lines of code (LOC), which the generated triton code and host launcher is around 1200 LOC. Even without including the static dispatcher and host launcher in count, the generated Triton kernel alone has over 400 LOC. And this is only partially generated content, because FlexLA also generates code for other parallel strategies.

Then we demonstrate the implementation of DeltaNet using FlexLA in Listing 3. Note that DeltaNet involves complex data dependencies where intermediate results computed in the chunk phase (specifically $U$ and $W$) are required in the merge phase. FlexLA introduces a primitive forge.cache_result to explicit mark these tensors. The compiler then automatically handles the memory layout and data movement to ensure these values are efficiently reused across phases without redundant re-computation or manual memory management.

## G.1 IMPLEMENTATION AND CODE GENERATION FOR SCALAR GLA

```python
def chunk_mode_scalar_gla(k: Tensor, v: Tensor, g: Tensor) -> Tensor:
    """k: [C, K], v: [C, V], g: [C]"""
    g_cumsum = g.cumsum(dim=0)
    g_cumsum_last = g.sum(dim=0)
    g_cumsum = (g_cumsum_last - g_cumsum).unsqueeze(0).exp()
    k_fp32 = k.permute([1, 0]).to(g.dtype)
    k_decay = (k_fp32 * g_cumsum).to(v.dtype)
    chunk_state = k_decay @ v
    return chunk_state

def decay_mode_scalar_gla(prev_s: Tensor, chunk_state: Tensor, g: Tensor)
        -> Tensor:
    """g: [C]"""
    g_sum = g.sum(dim=0).exp()
    return prev_s * g_sum + chunk_state

def merge_mode_scalar_gla(
    q: Tensor,
    k: Tensor,
    v: Tensor,
    g: Tensor,
    chunk_state: Tensor,
    scale: Tensor,
) -> Tensor:
    """q: [C, K], k: [C, K], v: [C, V], chunk_state: [K, V]"""
    g_cumsum = g.cumsum(0)
    chunk_state = chunk_state.to(q.dtype)
    p = (q @ k.T).tril(0)
    p = (p * (g_cumsum[..., None] - g_cumsum[None, ...]).exp()).to(v.
    dtype)
    return (p @ v + q @ chunk_state * g_cumsum.exp()[..., None]) * scale

CONST_H = ConstExpr("H", H)
CONST_K = ConstExpr("K", K)
CONST_V = ConstExpr("V", V)
meta = {
    "q": SymbTensor(["T", CONST_H, CONST_K], dtype=dtype),
    "k": SymbTensor(["T", CONST_H, CONST_K], dtype=dtype),
    "v": SymbTensor(["T", CONST_H, CONST_V], dtype=dtype),
    "g": SymbTensor(["T", CONST_H], dtype=torch.float32),
    "prev_s": SymbTensor(["NS", CONST_H, CONST_K, CONST_V], dtype=torch.
    float32),
    "chunk_state": SymbTensor(["NC", CONST_H, CONST_K, CONST_V], dtype=
    dtype),
    "scale": 1 / math.sqrt(K),
}
```

```
43  LinearAttention(
44      input_meta=meta,
45      sp_group=pg if args.sp else None,
46      enable_aot=args.aot,
47      code_dir=args.dir,
48      chunk_mode=chunk_mode_scalar_gla,
49      decay_mode=decay_mode_scalar_gla,
50      merge_mode=merge_mode_scalar_gla,
51  )
```

Listing 1: Implementation of Scalar GLA in Forge

```
1   fuse_chunk_decay_kernel_signature = (
2       "*bf16:16, "
3       "*bf16:16, "
4       "*fp32, "
5       "*bf16:16, "
6       "*bf16:16, "
7       "*fp32, "
8       "*i32, "
9       "i32, "
10      "i32, "
11      "*i32, "
12      "%USE_INITIAL_STATE, "
13      "%H, "
14      "%K, "
15      "%V, "
16      "%CHUNK, "
17      "%BLK_K, "
18      "%BLK_V, "
19      "%USE_TMA"
20  )
21
22
23  def get_fuse_chunk_decay_kernel_info(B: int, T: int, H: int, K: int, V:
        int):
24      """Static dispatcher for fuse_chunk_decay_kernel. Auto-generated."""
25      D = [K, V]
26      key = (B, T, H, D)
27      if key in (
28          (1, 4096, 4, [64, 64]),
29          (2, 1024, 4, [64, 64]),
30      ):
31          BLK_K = 32
32          BLK_V = 32
33          num_warps = 8
34          num_ctas = 1
35          num_stages = 3
36          maxnreg = None
37      elif key in (
38          (1, 1024, 1, [64, 64]),
39          (1, 1024, 1, [128, 128]),
40          (1, 1024, 4, [64, 64]),
41          (1, 1024, 4, [128, 128]),
42          (1, 1024, 8, [64, 64]),
43          (1, 1024, 8, [128, 128]),
44          (1, 1024, 16, [64, 64]),
45          (1, 1024, 32, [64, 64]),
46          (1, 2048, 1, [64, 64]),
47          (1, 2048, 1, [128, 128]),
48          (1, 2048, 4, [64, 64]),
49          (1, 2048, 4, [128, 128]),
50          (1, 2048, 8, [64, 64]),
51          (1, 2048, 8, [128, 128]),
52          (1, 2048, 16, [64, 64]),
```

```
53          (1, 2048, 32, [64, 64]),
54          (1, 4096, 1, [64, 64]),
55          (1, 4096, 1, [128, 128]),
56          (1, 4096, 4, [128, 128]),
57          (1, 4096, 8, [64, 64]),
58          (1, 4096, 8, [128, 128]),
59          (1, 4096, 16, [64, 64]),
60          (1, 4096, 32, [64, 64]),
61          (1, 8192, 1, [64, 64]),
62          (1, 8192, 1, [128, 128]),
63          (1, 8192, 4, [64, 64]),
64          (1, 8192, 4, [128, 128]),
65          (1, 8192, 8, [64, 64]),
66          (1, 8192, 8, [128, 128]),
67          (1, 8192, 16, [64, 64]),
68          (1, 8192, 32, [64, 64]),
69          (1, 16384, 1, [64, 64]),
70          (1, 16384, 1, [128, 128]),
71          (1, 16384, 4, [64, 64]),
72          (1, 16384, 4, [128, 128]),
73          (1, 16384, 8, [64, 64]),
74          (1, 16384, 8, [128, 128]),
75          (1, 16384, 16, [64, 64]),
76          (1, 16384, 32, [64, 64]),
77          (1, 32768, 1, [64, 64]),
78          (1, 32768, 1, [128, 128]),
79          (1, 32768, 4, [64, 64]),
80          (1, 32768, 4, [128, 128]),
81          (1, 32768, 8, [64, 64]),
82          (1, 32768, 8, [128, 128]),
83          (1, 32768, 16, [64, 64]),
84          (1, 32768, 32, [64, 64]),
85          (1, 65536, 1, [64, 64]),
86          (1, 65536, 1, [128, 128]),
87          (1, 65536, 4, [64, 64]),
88          (1, 65536, 4, [128, 128]),
89          (1, 65536, 8, [64, 64]),
90          (1, 65536, 8, [128, 128]),
91          (1, 65536, 16, [64, 64]),
92          (1, 65536, 32, [64, 64]),
93          (1, 131072, 1, [64, 64]),
94          (1, 131072, 1, [128, 128]),
95          (1, 131072, 4, [64, 64]),
96          (1, 131072, 4, [128, 128]),
97          (1, 131072, 8, [64, 64]),
98          (1, 131072, 8, [128, 128]),
99          (1, 131072, 16, [64, 64]),
100         (1, 131072, 32, [64, 64]),
101         (1, 262144, 1, [64, 64]),
102         (1, 262144, 1, [128, 128]),
103         (1, 262144, 4, [64, 64]),
104         (1, 262144, 4, [128, 128]),
105         (1, 262144, 8, [64, 64]),
106         (1, 262144, 8, [128, 128]),
107         (1, 262144, 16, [64, 64]),
108         (1, 262144, 32, [64, 64]),
109         (2, 1024, 1, [64, 64]),
110         (2, 1024, 1, [128, 128]),
111         (2, 1024, 4, [128, 128]),
112         (2, 1024, 8, [64, 64]),
113         (2, 1024, 16, [64, 64]),
114         (2, 2048, 1, [64, 64]),
115         (2, 2048, 1, [128, 128]),
116         (2, 2048, 4, [64, 64]),
117         (2, 2048, 4, [128, 128]),
```

```
118            (2, 2048, 8, [64, 64]),
119            (2, 2048, 16, [64, 64]),
120            (2, 4096, 1, [64, 64]),
121            (2, 4096, 1, [128, 128]),
122            (2, 4096, 4, [64, 64]),
123            (2, 4096, 4, [128, 128]),
124            (2, 4096, 8, [64, 64]),
125            (2, 4096, 16, [64, 64]),
126            (2, 8192, 1, [64, 64]),
127            (2, 8192, 1, [128, 128]),
128            (2, 8192, 4, [64, 64]),
129            (2, 8192, 4, [128, 128]),
130            (2, 8192, 8, [64, 64]),
131            (2, 8192, 16, [64, 64]),
132            (2, 16384, 1, [64, 64]),
133            (2, 16384, 1, [128, 128]),
134            (2, 16384, 4, [64, 64]),
135            (2, 16384, 4, [128, 128]),
136            (2, 16384, 8, [64, 64]),
137            (2, 16384, 16, [64, 64]),
138            (2, 32768, 1, [64, 64]),
139            (2, 32768, 1, [128, 128]),
140            (2, 32768, 4, [64, 64]),
141            (2, 32768, 4, [128, 128]),
142            (2, 32768, 8, [64, 64]),
143            (2, 32768, 16, [64, 64]),
144            (2, 65536, 1, [64, 64]),
145            (2, 65536, 1, [128, 128]),
146            (2, 65536, 4, [64, 64]),
147            (2, 65536, 4, [128, 128]),
148            (2, 65536, 8, [64, 64]),
149            (2, 65536, 16, [64, 64]),
150            (2, 131072, 1, [64, 64]),
151            (2, 131072, 1, [128, 128]),
152            (2, 131072, 4, [64, 64]),
153            (2, 131072, 4, [128, 128]),
154            (2, 131072, 8, [64, 64]),
155            (2, 131072, 16, [64, 64]),
156        ):
157            BLK_K = 32
158            BLK_V = 32
159            num_warps = 8
160            num_ctas = 1
161            num_stages = 4
162            maxnreg = None
163        elif key in (
164            (1, 1024, 32, [128, 128]),
165            (1, 2048, 32, [128, 128]),
166            (1, 4096, 32, [128, 128]),
167            (1, 8192, 32, [128, 128]),
168            (1, 16384, 32, [128, 128]),
169            (1, 32768, 32, [128, 128]),
170            (1, 65536, 32, [128, 128]),
171            (1, 131072, 32, [128, 128]),
172            (1, 262144, 32, [128, 128]),
173            (2, 1024, 32, [128, 128]),
174            (2, 2048, 16, [128, 128]),
175            (2, 2048, 32, [128, 128]),
176            (2, 4096, 16, [128, 128]),
177            (2, 4096, 32, [128, 128]),
178            (2, 8192, 32, [128, 128]),
179            (2, 16384, 16, [128, 128]),
180            (2, 16384, 32, [128, 128]),
181            (2, 32768, 32, [128, 128]),
182            (2, 131072, 16, [128, 128]),
```

```
183      ):
184          BLK_K = 32
185          BLK_V = 64
186          num_warps = 4
187          num_ctas = 1
188          num_stages = 3
189          maxnreg = None
190      elif key in (
191          (2, 1024, 16, [128, 128]),
192      ):
193          BLK_K = 32
194          BLK_V = 64
195          num_warps = 8
196          num_ctas = 1
197          num_stages = 3
198          maxnreg = None
199      elif key in (
200          (1, 1024, 16, [128, 128]),
201          (1, 2048, 16, [128, 128]),
202          (1, 4096, 16, [128, 128]),
203          (1, 8192, 16, [128, 128]),
204          (1, 16384, 16, [128, 128]),
205          (1, 32768, 16, [128, 128]),
206          (1, 65536, 16, [128, 128]),
207          (1, 131072, 16, [128, 128]),
208          (1, 262144, 16, [128, 128]),
209          (2, 1024, 8, [128, 128]),
210          (2, 1024, 32, [64, 64]),
211          (2, 2048, 8, [128, 128]),
212          (2, 2048, 32, [64, 64]),
213          (2, 4096, 8, [128, 128]),
214          (2, 4096, 32, [64, 64]),
215          (2, 8192, 8, [128, 128]),
216          (2, 8192, 16, [128, 128]),
217          (2, 8192, 32, [64, 64]),
218          (2, 16384, 8, [128, 128]),
219          (2, 16384, 32, [64, 64]),
220          (2, 32768, 8, [128, 128]),
221          (2, 32768, 16, [128, 128]),
222          (2, 32768, 32, [64, 64]),
223          (2, 65536, 8, [128, 128]),
224          (2, 65536, 16, [128, 128]),
225          (2, 65536, 32, [64, 64]),
226          (2, 131072, 8, [128, 128]),
227          (2, 131072, 32, [64, 64]),
228      ):
229          BLK_K = 32
230          BLK_V = 64
231          num_warps = 8
232          num_ctas = 1
233          num_stages = 4
234          maxnreg = None
235      elif key in (
236          (2, 65536, 32, [128, 128]),
237          (2, 131072, 32, [128, 128]),
238      ):
239          BLK_K = 64
240          BLK_V = 128
241          num_warps = 8
242          num_ctas = 1
243          num_stages = 4
244          maxnreg = None
245      else:
246          raise ValueError(f"Unsupported config for fuse_chunk_decay_kernel
         : BTHD={key}")
```

```
247
248     return {
249         "CHUNK": 64,
250         "USE_INITIAL_STATE": True,
251         "H": H,
252         "K": D[0],
253         "V": D[1],
254         "BLK_K": BLK_K,
255         "BLK_V": BLK_V,
256         "num_warps": num_warps,
257         "num_stages": num_stages,
258     }
259
260
261
262
263 @aot_compile_spaces({
264     "fuse_chunk_decay_kernel": {
265         "signature": fuse_chunk_decay_kernel_signature,
266         "grid": ["(%K + %BLK_K - 1) / %BLK_K", "(%V + %BLK_V - 1) / %
    BLK_V", "H_MUL_NS"],
267         "triton_algo_infos": [
268             get_fuse_chunk_decay_kernel_info(B, T, H, K, V)
269             for B,T,H,(K,V) in[
270                 (1, 262144, 32, [64, 64]),
271                 (1, 262144, 8, [128, 128]),
272                 (2, 1024, 16, [128, 128]),
273                 (2, 1024, 4, [64, 64]),
274                 (2, 131072, 1, [128, 128]),
275                 (2, 131072, 1, [64, 64]),
276                 (2, 131072, 16, [128, 128]),
277                 (2, 131072, 16, [64, 64]),
278                 (2, 131072, 32, [128, 128]),
279                 (2, 131072, 32, [64, 64]),
280                 (2, 131072, 4, [128, 128]),
281                 (2, 131072, 4, [64, 64]),
282                 (2, 131072, 8, [128, 128]),
283                 (2, 131072, 8, [64, 64]),
284                 (2, 32768, 32, [128, 128]),
285                 (2, 65536, 16, [128, 128])
286             ]
287         ],
288     }
289 })
290 @triton.autotune(
291     configs=[
292         triton.Config({"BLK_K": BLK_K, "BLK_V": BLK_V, "USE_TMA": USE_TMA
    }, num_warps=num_warps, num_stages=num_stages)
293         for num_warps in [4, 8]
294         for num_stages in [3, 4]
295         for BLK_K in [128, 64, 32]
296         for BLK_V in [128, 64, 32]
297         for USE_TMA in [True, False]
298     ],
299     key=[],
300 )
301 @triton.jit
302 def fuse_chunk_decay_kernel(
303     k,
304     v,
305     g,
306     prev_s,
307     out_0,
308     out_1,
309     cu_seqlens,
```

```
310      NS,
311      H_MUL_NS,
312      chunk_offsets_with_ini,
313      USE_INITIAL_STATE: tl.constexpr,
314      H: tl.constexpr,
315      K: tl.constexpr,
316      V: tl.constexpr,
317      CHUNK: tl.constexpr,
318      BLK_K: tl.constexpr,
319      BLK_V: tl.constexpr,
320      USE_TMA: tl.constexpr,
321 ):
322      NUM_BLK_K = (K + BLK_K - 1) // BLK_K
323      NUM_BLK_V = (V + BLK_V - 1) // BLK_V
324      NUM_BLK_KV = NUM_BLK_K * NUM_BLK_V
325
326      i_k, i_v, i_sh = tl.program_id(0), tl.program_id(1), tl.program_id(2)
327      i_s, i_h = i_sh // H, i_sh % H
328      bos, eos = tl.load(cu_seqlens + i_s).to(tl.int32), tl.load(cu_seqlens
          + i_s + 1).to(tl.int32)
329      T = eos - bos
330      NC = tl.cdiv(T, CHUNK)
331      boh = tl.load(chunk_offsets_with_ini + i_s).to(tl.int32)
332
333      out_0 = out_0 + (boh * H + i_h).to(tl.int64) * K * V
334      prev_s = prev_s + (i_s * H + i_h) * K * V
335      initial_decay = 1.0
336      # out_1: [NC + NS, H]. NOTE: **not** in log space
337      ptr_out_1 = out_1 + boh * H * i_h
338      # store initial decay
339      tl.store(ptr_out_1, initial_decay)
340      ptr_out_1 += H
341      # [BK, BV]
342      blk_prev_s = tl.zeros([BLK_K, BLK_V], dtype=tl.float32)
343      if USE_INITIAL_STATE:
344          ptr_prev_s = tl.make_block_ptr(prev_s, (K, V), (V, 1), (i_k *
          BLK_K, i_v * BLK_V), (BLK_K, BLK_V), (1, 0))
345          blk_prev_s = tl.load(ptr_prev_s, boundary_check=(0, 1)).to(tl.
          float32)
346      ptr_out_0 = tl.make_block_ptr(out_0, (K, V), (V, 1), (i_k*BLK_K, i_v*
          BLK_V), (BLK_K, BLK_V), (1, 0))  # fmt: skip
347      tl.store(ptr_out_0, blk_prev_s.to(ptr_out_0.dtype.element_ty))
348      out_0 += H * K * V
349
350      if USE_TMA:
351          k_desc = tl.make_tensor_descriptor(
352              k + bos * H * K + i_h * K,
353              shape=[T, K],
354              strides=[H * K, 1],
355              block_shape=[CHUNK, BLK_K],
356          )
357          v_desc = tl.make_tensor_descriptor(
358              v + bos * H * V + i_h * V,
359              shape=[T, V],
360              strides=[H * V, 1],
361              block_shape=[CHUNK, BLK_V],
362          )
363          out_0_desc = tl.make_tensor_descriptor(
364              out_0,
365              shape=[NC, K, V],
366              strides=[H * K * V, V, 1],
367              block_shape=[1, BLK_K, BLK_V],
368          )
369
370      for i_c in range(NC):
```

```
371         # load (trans) 'k': (T, H, K,) => (BLK_K, CHUNK,)
372         if not USE_TMA:
373             cur_k = k + bos * H * K + i_h * K  # fmt: skip
374             ptr_k_0 = tl.make_block_ptr(cur_k, (K, T,), (1, H * K,), (i_k
     * BLK_K, i_c * CHUNK,), (BLK_K, CHUNK,), (0, 1,))  # fmt: skip
375             blk_k_0 = tl.load(ptr_k_0, boundary_check=(0, 1,))  # fmt:
     skip
376         else:
377             blk_k_0 = k_desc.load([i_c * CHUNK, i_k * BLK_K]).trans()
378
379         # load 'v': (T, H, V,) => (CHUNK, BLK_V,)
380         if not USE_TMA:
381             cur_v = v + bos * H * V + i_h * V  # fmt: skip
382             ptr_v_1 = tl.make_block_ptr(cur_v, (T, V,), (H * V, 1,), (i_c
     * CHUNK, i_v * BLK_V,), (CHUNK, BLK_V,), (1, 0,))  # fmt: skip
383             blk_v_1 = tl.load(ptr_v_1, boundary_check=(1, 0,))  # fmt:
     skip
384         else:
385             blk_v_1 = v_desc.load([i_c * CHUNK, i_v * BLK_V])
386
387
388
389         # call_external_func: 'chunk_local_cumsum' to pre-compute g
390
391         # load 'g': (T, H,) => (CHUNK,)
392         cur_g = g + bos * H + i_h * 1  # fmt: skip
393         ptr_g_2 = tl.make_block_ptr(cur_g, (T,), (H,), (i_c * CHUNK,), (
     CHUNK,), (0,))  # fmt: skip
394         blk_g_2 = tl.load(ptr_g_2, boundary_check=(0,))  # fmt: skip
395         # load_last_g: => ()
396         # load 'g' [(CHUNK - 1)]: (T, H,) => (,)
397         cur_g = g + bos * H + i_h * 1 + (CHUNK - 1) * H  # fmt: skip
398         ptr_last_g_3 = cur_g + i_c * CHUNK * H  # fmt: skip
399         last_g_3 = tl.load(ptr_last_g_3)
400         # sub: torch.Size([]), ('CHUNK',) => ('CHUNK',)
401         sub_4 = last_g_3 - blk_g_2
402         # unsqueeze: ('CHUNK',) => (1, 'CHUNK')
403         unsqueeze_5 = sub_4[None, :]
404         # exp: (1, 'CHUNK') => (1, 'CHUNK')
405         exp_6 = tl.exp(unsqueeze_5)
406         # to: ('BLK_K', 'CHUNK') => ('BLK_K', 'CHUNK')
407         blk_k_0_float32_7 = blk_k_0.to(tl.float32)
408         # mul: ('BLK_K', 'CHUNK'), (1, 'CHUNK') => ('BLK_K', 'CHUNK')
409         mul_8 = blk_k_0_float32_7 * exp_6
410         # to: ('BLK_K', 'CHUNK') => ('BLK_K', 'CHUNK')
411         mul_8_bfloat16_9 = mul_8.to(tl.bfloat16)
412         # matmul: ('BLK_K', 'CHUNK'), ('CHUNK', 'BLK_V') => ('BLK_K', '
     BLK_V')
413         matmul_10 = tl.dot(mul_8_bfloat16_9, blk_v_1).to(mul_8_bfloat16_9
     .dtype)
414         # exp: () => ()
415         exp_1_11 = tl.exp(last_g_3)
416         # mul: ('BLK_K', 'BLK_V'), () => ('BLK_K', 'BLK_V')
417         mul_1_12 = blk_prev_s * exp_1_11
418         # add: ('BLK_K', 'BLK_V'), ('BLK_K', 'BLK_V') => ('BLK_K', 'BLK_V
     ')
419         add_13 = mul_1_12 + matmul_10
420         # mul: ('s3',), () => ('s3',)
421         mul_tensor_14 = initial_decay * exp_1_11
422         # store => ('BLK_K', 'BLK_V')
423         # assume output layout: [NC_WITH_INI, H, K, V]
424         out_0_ty = out_0.dtype.element_ty
425         if not USE_TMA:
426             ptr_out_0 = tl.make_block_ptr(out_0, (K, V,), (V, 1,), (i_k *
     BLK_K, i_v * BLK_V,), (BLK_K, BLK_V,), (1, 0))  # fmt: skip
```

```
427            tl.store(ptr_out_0, add_13.to(out_0_ty), boundary_check=(1,
     0))
428            out_0 += H * K * V
429        else:
430            out_0_desc.store([i_c, i_k * BLK_K, i_v * BLK_V], add_13.to(
     out_0_ty)[None, :, :])
431        blk_prev_s = add_13.to(blk_prev_s.dtype)
432        # store => ('s3',)
433        # output layout: [NC_WITH_INI, H, s3]
434        out_1_ty = out_1.dtype.element_ty
435        tl.store(ptr_out_1 + i_c * H, mul_tensor_14.to(out_1_ty))
436        initial_decay = mul_tensor_14

438 def launch_fuse_chunk_decay(
439     g: torch.Tensor,
440     prev_s: torch.Tensor,
441     v: torch.Tensor,
442     k: torch.Tensor,
443     cu_seqlens: torch.IntTensor,
444     cached_results: dict,
445     dist_scan: DistScanContext = None,
446     lazy_update: bool = True,
447 ) -> dict[str, torch.Tensor]:
448     '''
449     perform "decayed scan" on `chunked_states`
450     '''
451     # state_dtype = chunk_state.dtype
452     chunk_decay = None
453     prev_rank_state_sum = None
454
455     # TODO: now fix chunk_size to 64
456     CHUNK = 64
457     chunk_indices = prepare_chunk_indices(cu_seqlens, CHUNK)
458     chunk_offsets = prepare_chunk_offsets(cu_seqlens, CHUNK)
459     chunk_offsets_with_ini = prepare_chunk_offsets_with_ini(cu_seqlens,
     CHUNK)
460     NS, NC = len(cu_seqlens) - 1, chunk_indices.shape[0]
461     NC_WITH_INI = NC + NS
462     use_aot = os.environ.get('FORGE_USE_AOT', '0')
463     FORGE_USE_AOT = True if use_aot.lower() in ['1', 'true', 'yes'] else
     None
464     T, H, K, V = *k.shape, v.shape[-1]
465     updated_states = k.new_empty([NC_WITH_INI, H, K, V])
466
467     def alloc_fn(size: int, alignment: int, stream: int):
468         return torch.empty(size, device='cuda', dtype=torch.int8)
469
470     triton.set_allocator(alloc_fn)
471
472     def grid(meta):
473         BLK_K = meta['BLK_K']
474         BLK_V = meta['BLK_V']
475         NUM_BLK_K = (K + BLK_K - 1) // BLK_K
476         NUM_BLK_V = (V + BLK_V - 1) // BLK_V
477         NUM_BLK_KV = NUM_BLK_K * NUM_BLK_V
478         H_MUL_NS = H * NS
479         return (NUM_BLK_K, NUM_BLK_V, H_MUL_NS,)
480
481     chunk_decay = torch.empty([NC_WITH_INI, H], dtype=torch.float32)
482     if FORGE_USE_AOT is None:
483         fuse_chunk_decay_kernel[grid](
484             g=g,
485             prev_s=prev_s,
486             v=v,
487             k=k,
```

```
488                out_1=chunk_decay,
489                cu_seqlens=cu_seqlens,
490                H=H,
491                NS=NS,
492                K=K,
493                V=V,
494                CHUNK=CHUNK,
495                H_MUL_NS=H * NS,
496                out_0=updated_states,
497                chunk_offsets_with_ini=chunk_offsets_with_ini,
498                USE_INITIAL_STATE=True,
499            )
500
501        else:
502
503            from foge.aot_utils import forge_aot_ops
504
505            algo_info = forge_aot_ops.
      fuse_chunk_decay_kernel__triton_algo_info_t()
506            for _k, _v in get_fuse_chunk_decay_kernel_info(NS, T, H, K, V).
      items():
507                setattr(algo_info, _k, _v)
508            forge_aot_ops.fuse_chunk_decay_kernel(
509                0,  # torch.cuda.current_stream().cuda_stream,
510                k.data_ptr(),  # k
511                v.data_ptr(),  # v
512                g.data_ptr(),  # g
513                prev_s.data_ptr(),  # prev_s
514                updated_states.data_ptr(),  # out_0
515                chunk_decay.data_ptr(),  # out_1
516                cu_seqlens.data_ptr(),  # cu_seqlens
517                NS,  # NS
518                H * NS,  # H_MUL_NS
519                chunk_offsets_with_ini.data_ptr(),  # chunk_offsets_with_ini
520                algo_info,
521            )
522
523
524        final_chunk_indices = chunk_offsets_with_ini[1:] - 1
525        final_state_local = updated_states[final_chunk_indices, ...]
526        final_decay_local = chunk_decay[final_chunk_indices, ...]
527
528        if dist_scan.pg.size() > 1:
529            prev_rank_state_sum = dist_scan.forward(
530                final_state_local=final_state_local,
531                final_decay_local=final_decay_local,
532                decay_type=DecayType.SCALAR,
533                lazy_update=True,
534            )
535
536        cached_results.update({})
537        if prev_rank_state_sum is not None:
538            return {
539                "chunk_state": updated_states,
540                "chunk_decay": chunk_decay,
541                "prev_rank_state_sum": prev_rank_state_sum,
542            }
543        else:
544            return {
545                "chunk_state": updated_states,
546                "chunk_decay": None,
547                "prev_rank_state_sum": None,
548            }
549 merge_mode_kernel_signature = (
550    "*bf16:16, "
```

```
551         "*bf16:16, "
552         "*bf16:16, "
553         "*fp32, "
554         "*bf16:16, "
555         "fp32, "
556         "*bf16:16, "
557         "i32:16, "
558         "*bf16:16, "
559         "i32:16, "
560         "*bf16:16, "
561         "*i32, "
562         "i32, "
563         "i32, "
564         "*i32, "
565         "i32, "
566         "%FUSE_SP_STATE_UPDATE, "
567         "%H, "
568         "%K, "
569         "%V, "
570         "%CHUNK, "
571         "%BLK_K, "
572         "%BLK_V, "
573         "%USE_TMA"
574     )
575
576
577 def get_merge_mode_kernel_info(B: int, T: int, H: int, K: int, V: int):
578     """Static dispatcher for merge_mode_kernel. Auto-generated."""
579     D = [K, V]
580     key = (B, T, H, D)
581     if key in (
582         (1, 4096, 4, [128, 128]),
583         (1, 131072, 32, [128, 128]),
584         (1, 262144, 32, [128, 128]),
585         (2, 1024, 8, [128, 128]),
586         (2, 4096, 1, [128, 128]),
587         (2, 8192, 1, [128, 128]),
588     ):
589         BLK_K = 128
590         BLK_V = 128
591         num_warps = 4
592         num_ctas = 1
593         num_stages = 3
594         maxnreg = None
595     elif key in (
596         (1, 4096, 32, [128, 128]),
597         (1, 16384, 8, [128, 128]),
598         (1, 16384, 32, [128, 128]),
599         (1, 32768, 8, [128, 128]),
600         (1, 32768, 16, [128, 128]),
601         (1, 32768, 32, [128, 128]),
602         (1, 65536, 8, [128, 128]),
603         (1, 65536, 32, [128, 128]),
604         (1, 131072, 4, [128, 128]),
605         (1, 131072, 8, [128, 128]),
606         (1, 131072, 16, [128, 128]),
607         (1, 262144, 4, [128, 128]),
608         (1, 262144, 8, [128, 128]),
609         (1, 262144, 16, [128, 128]),
610         (2, 4096, 16, [128, 128]),
611         (2, 4096, 32, [128, 128]),
612         (2, 16384, 8, [128, 128]),
613         (2, 16384, 32, [128, 128]),
614         (2, 32768, 16, [128, 128]),
615         (2, 32768, 32, [128, 128]),
```

```
616            (2, 65536, 8, [128, 128]),
617            (2, 65536, 16, [128, 128]),
618            (2, 131072, 4, [128, 128]),
619            (2, 131072, 8, [128, 128]),
620            (2, 131072, 16, [128, 128]),
621        ):
622            BLK_K = 128
623            BLK_V = 128
624            num_warps = 8
625            num_ctas = 1
626            num_stages = 3
627            maxnreg = None
628        elif key in (
629            (1, 1024, 16, [128, 128]),
630            (1, 1024, 32, [128, 128]),
631            (1, 2048, 8, [128, 128]),
632            (1, 2048, 16, [128, 128]),
633            (1, 16384, 1, [128, 128]),
634            (1, 32768, 1, [128, 128]),
635            (2, 2048, 4, [128, 128]),
636            (2, 65536, 32, [128, 128]),
637            (2, 131072, 32, [128, 128]),
638        ):
639            BLK_K = 128
640            BLK_V = 128
641            num_warps = 4
642            num_ctas = 1
643            num_stages = 4
644            maxnreg = None
645        elif key in (
646            (1, 2048, 32, [128, 128]),
647            (1, 4096, 16, [128, 128]),
648            (1, 8192, 16, [128, 128]),
649            (1, 8192, 32, [128, 128]),
650            (1, 16384, 16, [128, 128]),
651            (1, 65536, 4, [128, 128]),
652            (1, 65536, 16, [128, 128]),
653            (2, 1024, 32, [128, 128]),
654            (2, 2048, 32, [128, 128]),
655            (2, 8192, 8, [128, 128]),
656            (2, 8192, 16, [128, 128]),
657            (2, 8192, 32, [128, 128]),
658            (2, 16384, 16, [128, 128]),
659            (2, 32768, 4, [128, 128]),
660            (2, 32768, 8, [128, 128]),
661            (2, 65536, 4, [128, 128]),
662        ):
663            BLK_K = 128
664            BLK_V = 128
665            num_warps = 8
666            num_ctas = 1
667            num_stages = 4
668            maxnreg = None
669        elif key in (
670            (1, 1024, 1, [64, 64]),
671            (1, 4096, 1, [64, 64]),
672        ):
673            BLK_K = 128
674            BLK_V = 32
675            num_warps = 4
676            num_ctas = 1
677            num_stages = 3
678            maxnreg = None
679        elif key in (
680            (1, 1024, 1, [128, 128]),
```

```
681            (1, 1024, 4, [64, 64]),
682            (1, 1024, 8, [64, 64]),
683            (1, 2048, 1, [64, 64]),
684            (1, 2048, 1, [128, 128]),
685            (1, 2048, 4, [64, 64]),
686            (2, 1024, 1, [64, 64]),
687            (2, 2048, 1, [64, 64]),
688        ):
689            BLK_K = 128
690            BLK_V = 32
691            num_warps = 4
692            num_ctas = 1
693            num_stages = 4
694            maxnreg = None
695        elif key in (
696            (1, 1024, 8, [128, 128]),
697            (1, 1024, 32, [64, 64]),
698            (1, 2048, 4, [128, 128]),
699            (1, 4096, 4, [64, 64]),
700            (1, 4096, 8, [64, 64]),
701            (1, 4096, 8, [128, 128]),
702            (1, 8192, 1, [128, 128]),
703            (1, 8192, 8, [64, 64]),
704            (1, 8192, 8, [128, 128]),
705            (1, 8192, 32, [64, 64]),
706            (1, 16384, 1, [64, 64]),
707            (1, 16384, 32, [64, 64]),
708            (1, 32768, 1, [64, 64]),
709            (1, 32768, 4, [64, 64]),
710            (1, 32768, 8, [64, 64]),
711            (1, 32768, 16, [64, 64]),
712            (1, 65536, 1, [128, 128]),
713            (1, 131072, 1, [128, 128]),
714            (1, 131072, 32, [64, 64]),
715            (1, 262144, 1, [64, 64]),
716            (1, 262144, 32, [64, 64]),
717            (2, 1024, 4, [64, 64]),
718            (2, 1024, 4, [128, 128]),
719            (2, 1024, 8, [64, 64]),
720            (2, 1024, 16, [64, 64]),
721            (2, 1024, 32, [64, 64]),
722            (2, 2048, 4, [64, 64]),
723            (2, 2048, 8, [64, 64]),
724            (2, 2048, 8, [128, 128]),
725            (2, 2048, 16, [64, 64]),
726            (2, 2048, 16, [128, 128]),
727            (2, 2048, 32, [64, 64]),
728            (2, 4096, 4, [64, 64]),
729            (2, 4096, 4, [128, 128]),
730            (2, 4096, 8, [64, 64]),
731            (2, 4096, 8, [128, 128]),
732            (2, 4096, 16, [64, 64]),
733            (2, 4096, 32, [64, 64]),
734            (2, 8192, 1, [64, 64]),
735            (2, 8192, 4, [128, 128]),
736            (2, 16384, 1, [64, 64]),
737            (2, 16384, 1, [128, 128]),
738            (2, 16384, 4, [128, 128]),
739            (2, 16384, 8, [64, 64]),
740            (2, 16384, 32, [64, 64]),
741            (2, 32768, 16, [64, 64]),
742            (2, 32768, 32, [64, 64]),
743            (2, 65536, 1, [64, 64]),
744            (2, 65536, 1, [128, 128]),
745            (2, 65536, 16, [64, 64]),
```

```
746    ):
747        BLK_K = 128
748        BLK_V = 64
749        num_warps = 4
750        num_ctas = 1
751        num_stages = 3
752        maxnreg = None
753    elif key in (
754        (1, 4096, 1, [128, 128]),
755    ):
756        BLK_K = 128
757        BLK_V = 64
758        num_warps = 8
759        num_ctas = 1
760        num_stages = 3
761        maxnreg = None
762    elif key in (
763        (1, 1024, 4, [128, 128]),
764        (1, 1024, 16, [64, 64]),
765        (1, 2048, 8, [64, 64]),
766        (1, 2048, 16, [64, 64]),
767        (1, 2048, 32, [64, 64]),
768        (1, 4096, 16, [64, 64]),
769        (1, 4096, 32, [64, 64]),
770        (1, 8192, 1, [64, 64]),
771        (1, 8192, 4, [64, 64]),
772        (1, 8192, 4, [128, 128]),
773        (1, 8192, 16, [64, 64]),
774        (1, 16384, 4, [64, 64]),
775        (1, 16384, 4, [128, 128]),
776        (1, 16384, 8, [64, 64]),
777        (1, 16384, 16, [64, 64]),
778        (1, 32768, 4, [128, 128]),
779        (1, 32768, 32, [64, 64]),
780        (1, 65536, 1, [64, 64]),
781        (1, 65536, 4, [64, 64]),
782        (1, 65536, 8, [64, 64]),
783        (1, 65536, 16, [64, 64]),
784        (1, 65536, 32, [64, 64]),
785        (1, 131072, 1, [64, 64]),
786        (1, 131072, 4, [64, 64]),
787        (1, 131072, 8, [64, 64]),
788        (1, 131072, 16, [64, 64]),
789        (1, 262144, 1, [128, 128]),
790        (1, 262144, 4, [64, 64]),
791        (1, 262144, 8, [64, 64]),
792        (1, 262144, 16, [64, 64]),
793        (2, 1024, 16, [128, 128]),
794        (2, 8192, 4, [64, 64]),
795        (2, 8192, 8, [64, 64]),
796        (2, 8192, 16, [64, 64]),
797        (2, 8192, 32, [64, 64]),
798        (2, 16384, 4, [64, 64]),
799        (2, 16384, 16, [64, 64]),
800        (2, 32768, 1, [64, 64]),
801        (2, 32768, 1, [128, 128]),
802        (2, 32768, 4, [64, 64]),
803        (2, 32768, 8, [64, 64]),
804        (2, 65536, 4, [64, 64]),
805        (2, 65536, 8, [64, 64]),
806        (2, 65536, 32, [64, 64]),
807        (2, 131072, 1, [64, 64]),
808        (2, 131072, 1, [128, 128]),
809        (2, 131072, 4, [64, 64]),
810        (2, 131072, 8, [64, 64]),
```

```
811            (2, 131072, 16, [64, 64]),
812            (2, 131072, 32, [64, 64]),
813        ):
814            BLK_K = 128
815            BLK_V = 64
816            num_warps = 4
817            num_ctas = 1
818            num_stages = 4
819            maxnreg = None
820        elif key in (
821            (2, 1024, 1, [128, 128]),
822            (2, 2048, 1, [128, 128]),
823            (2, 4096, 1, [64, 64]),
824        ):
825            BLK_K = 128
826            BLK_V = 64
827            num_warps = 8
828            num_ctas = 1
829            num_stages = 4
830            maxnreg = None
831        else:
832            raise ValueError(f"Unsupported config for merge_mode_kernel: BTHD
    ={key}")
833
834        return {
835            "CHUNK": 64,
836            "FUSE_SP_STATE_UPDATE": True,
837            "H": H,
838            "K": D[0],
839            "V": D[1],
840            "BLK_K": BLK_K,
841            "BLK_V": BLK_V,
842            "num_warps": num_warps,
843            "num_stages": num_stages,
844        }
845
846
847
848
849 @aot_compile_spaces({
850     "merge_mode_kernel": {
851         "signature": merge_mode_kernel_signature,
852         "grid": ["((%K + %BLK_K - 1) / %BLK_K) * ((%V + %BLK_V - 1) / %
    BLK_V)", "%H", "NC"],
853         "triton_algo_infos": [
854             get_merge_mode_kernel_info(B, T, H, K, V)
855             for B,T,H,(K,V) in[
856                 (1, 1024, 8, [64, 64]),
857                 (1, 2048, 1, [128, 128]),
858                 (1, 2048, 16, [128, 128]),
859                 (1, 2048, 4, [64, 64]),
860                 (1, 2048, 8, [128, 128]),
861                 (1, 262144, 32, [128, 128]),
862                 (1, 32768, 1, [128, 128]),
863                 (1, 32768, 4, [128, 128]),
864                 (1, 4096, 1, [128, 128]),
865                 (1, 4096, 1, [64, 64]),
866                 (1, 4096, 4, [128, 128]),
867                 (2, 1024, 16, [128, 128]),
868                 (2, 1024, 8, [128, 128]),
869                 (2, 131072, 1, [128, 128]),
870                 (2, 131072, 1, [64, 64]),
871                 (2, 131072, 16, [128, 128]),
872                 (2, 131072, 16, [64, 64]),
873                 (2, 131072, 32, [128, 128]),
```

```
874                    (2, 131072, 32, [64, 64]),
875                    (2, 131072, 4, [128, 128]),
876                    (2, 131072, 4, [64, 64]),
877                    (2, 131072, 8, [128, 128]),
878                    (2, 131072, 8, [64, 64]),
879                    (2, 16384, 16, [128, 128]),
880                    (2, 16384, 4, [128, 128]),
881                    (2, 16384, 8, [64, 64]),
882                    (2, 2048, 1, [128, 128]),
883                    (2, 2048, 1, [64, 64]),
884                    (2, 2048, 16, [128, 128]),
885                    (2, 2048, 4, [128, 128]),
886                    (2, 32768, 32, [128, 128]),
887                    (2, 32768, 32, [64, 64]),
888                    (2, 32768, 8, [128, 128]),
889                    (2, 4096, 1, [64, 64]),
890                    (2, 4096, 4, [64, 64]),
891                    (2, 4096, 8, [128, 128]),
892                    (2, 65536, 1, [128, 128]),
893                    (2, 65536, 1, [64, 64]),
894                    (2, 65536, 16, [64, 64]),
895                    (2, 65536, 4, [128, 128]),
896                    (2, 8192, 1, [128, 128]),
897                    (2, 8192, 32, [128, 128])
898                ]
899            ],
900        }
901 })
902 @triton.autotune(
903     configs=[
904         triton.Config({"BLK_K": BLK_K, "BLK_V": BLK_V, "USE_TMA": USE_TMA
     }, num_warps=num_warps, num_stages=num_stages)
905         for num_warps in [4, 8]
906         for num_stages in [3, 4]
907         for BLK_K in [128]
908         for BLK_V in [128, 64, 32]
909         for USE_TMA in [True, False]
910     ],
911     key=[],
912 )
913 @triton.jit
914 def merge_mode_kernel(
915     q,
916     k,
917     v,
918     g,
919     chunk_state,
920     scale,
921     prev_rank_state_sum,
922     stride_d0_ns_prev_rank_state_sum,
923     chunk_decay,
924     stride_d0_nc_with_ini_chunk_decay,
925     out_0,
926     cu_seqlens,
927     NS,
928     H_MUL_NS,
929     chunk_indices,
930     NC,
931     FUSE_SP_STATE_UPDATE: tl.constexpr,
932     H: tl.constexpr,
933     K: tl.constexpr,
934     V: tl.constexpr,
935     CHUNK: tl.constexpr,
936     BLK_K: tl.constexpr,
937     BLK_V: tl.constexpr,
```

```
938      USE_TMA: tl.constexpr,
939  ):
940      NUM_BLK_K = (K + BLK_K - 1) // BLK_K
941      NUM_BLK_V = (V + BLK_V - 1) // BLK_V
942      NUM_BLK_KV = NUM_BLK_K * NUM_BLK_V
943
944      i_v, i_h, i_gc = tl.program_id(0), tl.program_id(1), tl.program_id(2)
945      i_s, i_c = tl.load(chunk_indices + i_gc * 2).to(tl.int32), tl.load(
         chunk_indices + i_gc * 2 + 1).to(tl.int32)
946      bos, eos = tl.load(cu_seqlens + i_s).to(tl.int32), tl.load(cu_seqlens
          + i_s + 1).to(tl.int32)
947      T = eos - bos
948      # NC = tl.cdiv(T, CHUNK)
949
950      # FIXME: set `i_k` temporarily
951      i_k = 0
952      out_0 = out_0 + (bos * H + i_h) * V
953
954      if USE_TMA:
955          q_desc = tl.make_tensor_descriptor(
956              q + bos * H * K + i_h * K,
957              shape=[T, K],
958              strides=[H * K, 1],
959              block_shape=[CHUNK, BLK_K],
960          )
961          k_desc = tl.make_tensor_descriptor(
962              k + bos * H * K + i_h * K,
963              shape=[T, K],
964              strides=[H * K, 1],
965              block_shape=[CHUNK, BLK_K],
966          )
967          v_desc = tl.make_tensor_descriptor(
968              v + bos * H * V + i_h * V,
969              shape=[T, V],
970              strides=[H * V, 1],
971              block_shape=[CHUNK, BLK_V],
972          )
973          chunk_state_desc = tl.make_tensor_descriptor(
974              chunk_state + (i_gc + i_s).to(tl.int64) * H * K * V + i_h * K
      * V,
975              shape=[K, V],
976              strides=[V, 1],
977              block_shape=[BLK_K, BLK_V],
978          )
979
980      if FUSE_SP_STATE_UPDATE:
981          prev_rank_state_sum_desc = tl.make_tensor_descriptor(
982              prev_rank_state_sum + i_s * stride_d0_ns_prev_rank_state_sum
      + i_h * K * V,
983              shape=[K, V],
984              strides=[V, 1],
985              block_shape=[BLK_K, BLK_V],
986          )
987
988      # load `q`: (T, H, K,) => (CHUNK, BLK_K,)
989      if not USE_TMA:
990          cur_q = q + bos * H * K + i_h * K  # fmt: skip
991          ptr_q_0 = tl.make_block_ptr(cur_q, (T, K,), (H * K, 1,), (i_c *
      CHUNK, i_k * BLK_K,), (CHUNK, BLK_K,), (1, 0,))  # fmt: skip
992          blk_q_0 = tl.load(ptr_q_0, boundary_check=(1, 0,))  # fmt: skip
993      else:
994          blk_q_0 = q_desc.load([i_c * CHUNK, i_k * BLK_K])
995
996      # load (trans) `k`: (T, H, K,) => (BLK_K, CHUNK,)
997      if not USE_TMA:
```

```
998             cur_k = k + bos * H * K + i_h * K  # fmt: skip
999             ptr_k_1 = tl.make_block_ptr(cur_k, (K, T,), (1, H * K,), (i_k *
        BLK_K, i_c * CHUNK,), (BLK_K, CHUNK,), (0, 1,))  # fmt: skip
1000            blk_k_1 = tl.load(ptr_k_1, boundary_check=(0, 1,))  # fmt: skip
1001        else:
1002            blk_k_1 = k_desc.load([i_c * CHUNK, i_k * BLK_K]).trans()
1003
1004        # load 'v': (T, H, V,) => (CHUNK, BLK_V,)
1005        if not USE_TMA:
1006            cur_v = v + bos * H * V + i_h * V  # fmt: skip
1007            ptr_v_2 = tl.make_block_ptr(cur_v, (T, V,), (H * V, 1,), (i_c *
        CHUNK, i_v * BLK_V,), (CHUNK, BLK_V,), (1, 0,))  # fmt: skip
1008            blk_v_2 = tl.load(ptr_v_2, boundary_check=(1, 0,))  # fmt: skip
1009        else:
1010            blk_v_2 = v_desc.load([i_c * CHUNK, i_v * BLK_V])
1011
1012        # call_external_func: 'chunk_local_cumsum' to pre-compute g
1013
1014        # load 'g': (T, H,) => (CHUNK,)
1015        cur_g = g + bos * H + i_h * 1  # fmt: skip
1016        ptr_g_3 = tl.make_block_ptr(cur_g, (T,), (H,), (i_c * CHUNK,), (CHUNK
        ,), (0,))  # fmt: skip
1017        blk_g_3 = tl.load(ptr_g_3, boundary_check=(0,))  # fmt: skip
1018        # load 'chunk_state': (NC_WITH_INI, H, K, V,) => (BLK_K, BLK_V,)
1019        if not USE_TMA:
1020            cur_chunk_state = chunk_state + (i_gc + i_s).to(tl.int64) * H * K
         * V + i_h * K * V  # fmt: skip
1021            ptr_chunk_state_4 = tl.make_block_ptr(cur_chunk_state, (K, V,), (
        V, 1,), (i_k * BLK_K, i_v * BLK_V,), (BLK_K, BLK_V,), (1, 0,))  # fmt
        : skip
1022            blk_chunk_state_4 = tl.load(ptr_chunk_state_4, boundary_check=(1,
         0,))  # fmt: skip
1023        else:
1024            blk_chunk_state_4 = chunk_state_desc.load([i_k * BLK_K, i_v *
        BLK_V])
1025
1026        # comments not available for op 'if_beg'
1027        if FUSE_SP_STATE_UPDATE:
1028            # load 'prev_rank_state_sum': (NS, H, K, V,) => (BLK_K, BLK_V,)
1029            if not USE_TMA:
1030                cur_prev_rank_state_sum = prev_rank_state_sum + i_s *
        stride_d0_ns_prev_rank_state_sum + i_h * K * V  # fmt: skip
1031                ptr_prev_rank_state_sum_5 = tl.make_block_ptr(
        cur_prev_rank_state_sum, (K, V,), (V, 1,), (i_k * BLK_K, i_v * BLK_V
        ,), (BLK_K, BLK_V,), (1, 0,))  # fmt: skip
1032                blk_prev_rank_state_sum_5 = tl.load(ptr_prev_rank_state_sum_5
        , boundary_check=(1, 0,))  # fmt: skip
1033            else:
1034                blk_prev_rank_state_sum_5 = prev_rank_state_sum_desc.load([
        i_k * BLK_K, i_v * BLK_V])
1035
1036            # load 'chunk_decay': (NC_WITH_INI, H,) => (,)
1037            cur_chunk_decay = chunk_decay + (i_gc + i_s).to(tl.int64) *
        stride_d0_nc_with_ini_chunk_decay + i_h * 1  # fmt: skip
1038            ptr_chunk_decay_6 = cur_chunk_decay  # fmt: skip
1039            blk_chunk_decay_6 = tl.load(ptr_chunk_decay_6)
1040            # to: ('BLK_K', 'BLK_V') => ('BLK_K', 'BLK_V')
1041            blk_prev_rank_state_sum_5_float32_7 = blk_prev_rank_state_sum_5.
        to(tl.float32)
1042            # mul: (), ('BLK_K', 'BLK_V') => ('BLK_K', 'BLK_V')
1043            mul_tensor_8 = blk_chunk_decay_6 *
        blk_prev_rank_state_sum_5_float32_7
1044            # to: ('BLK_K', 'BLK_V') => ('BLK_K', 'BLK_V')
1045            mul_tensor_8_bfloat16_9 = mul_tensor_8.to(tl.bfloat16)
```

```
1046          # add: ('BLK_K', 'BLK_V'), ('BLK_K', 'BLK_V') => ('BLK_K', 'BLK_V
         ')
1047          add_tensor_10 = mul_tensor_8_bfloat16_9 + blk_chunk_state_4
1048          # comments not available for op 'bind_var'
1049          blk_chunk_state_4 = add_tensor_10
1050          # comments not available for op 'end_if'
1051      # to: ('BLK_K', 'BLK_V') => ('BLK_K', 'BLK_V')
1052      blk_chunk_state_4_bfloat16_11 = blk_chunk_state_4.to(tl.bfloat16)
1053      # matmul: ('CHUNK', 'BLK_K'), ('CHUNK', 'BLK_K') => ('CHUNK', 'CHUNK
         ')
1054      matmul_12 = tl.dot(blk_q_0, blk_k_1).to(blk_q_0.dtype)
1055      # tril: ('CHUNK', 'CHUNK') => ('CHUNK', 'CHUNK')
1056      tril_13 = tl.where(tl.arange(0, CHUNK)[:, None] >= tl.arange(0, CHUNK
         )[None, :], matmul_12, 0)
1057      # unsqueeze: ('CHUNK',) => ('CHUNK', 1)
1058      unsqueeze_14 = blk_g_3[:, None]
1059      # unsqueeze: ('CHUNK',) => (1, 'CHUNK')
1060      unsqueeze_1_15 = blk_g_3[None, :]
1061      # sub: ('CHUNK', 1), (1, 'CHUNK') => ('CHUNK', 'CHUNK')
1062      sub_16 = unsqueeze_14 - unsqueeze_1_15
1063      # exp: ('CHUNK', 'CHUNK') => ('CHUNK', 'CHUNK')
1064      exp_17 = tl.exp(sub_16)
1065      # mul: ('CHUNK', 'CHUNK'), ('CHUNK', 'CHUNK') => ('CHUNK', 'CHUNK')
1066      mul_18 = tril_13 * exp_17
1067      # to: ('CHUNK', 'CHUNK') => ('CHUNK', 'CHUNK')
1068      mul_18_bfloat16_19 = mul_18.to(tl.bfloat16)
1069      # matmul: ('CHUNK', 'CHUNK'), ('CHUNK', 'BLK_V') => ('CHUNK', 'BLK_V
         ')
1070      matmul_1_20 = tl.dot(mul_18_bfloat16_19, blk_v_2).to(
         mul_18_bfloat16_19.dtype)
1071      # matmul: ('CHUNK', 'BLK_K'), ('BLK_K', 'BLK_V') => ('CHUNK', 'BLK_V
         ')
1072      matmul_2_21 = tl.dot(blk_q_0, blk_chunk_state_4_bfloat16_11).to(
         blk_q_0.dtype)
1073      # exp: ('CHUNK',) => ('CHUNK',)
1074      exp_1_22 = tl.exp(blk_g_3)
1075      # unsqueeze: ('CHUNK',) => ('CHUNK', 1)
1076      unsqueeze_2_23 = exp_1_22[:, None]
1077      # mul: ('CHUNK', 'BLK_V'), ('CHUNK', 1) => ('CHUNK', 'BLK_V')
1078      mul_1_24 = matmul_2_21 * unsqueeze_2_23
1079      # add: ('CHUNK', 'BLK_V'), ('CHUNK', 'BLK_V') => ('CHUNK', 'BLK_V')
1080      add_25 = matmul_1_20 + mul_1_24
1081      # mul: ('CHUNK', 'BLK_V'), () => ('CHUNK', 'BLK_V')
1082      mul_2_26 = add_25 * scale
1083      # store => ('CHUNK', 'BLK_V')
1084      # assume output layout: [T, H, V]
1085      out_0_ty = out_0.dtype.element_ty
1086      ptr_out_0 = tl.make_block_ptr(out_0, (T, V,), (H * V, 1,), (i_c *
         CHUNK, i_v * BLK_V,), (CHUNK, BLK_V,), (1, 0))  # fmt: skip
1087      tl.store(ptr_out_0, mul_2_26.to(out_0_ty), boundary_check=(1, 0))

1089 def launch_merge_mode(
1090      chunk_decay: torch.Tensor,
1091      v: torch.Tensor,
1092      k: torch.Tensor,
1093      q: torch.Tensor,
1094      g: torch.Tensor,
1095      prev_rank_state_sum: torch.Tensor,
1096      chunk_state: torch.Tensor,
1097      scale,
1098      cu_seqlens: torch.IntTensor,
1099      cached_results: dict,
1100      fuse_sp_update: bool,
1101 ) -> torch.Tensor:
1102      T, H, K, V = *k.shape, v.shape[-1]
```

```
1103
1104    if scale is None:
1105        scale = k.shape[-1] ** -0.5
1106
1107    # TODO: now fix chunk_size to 64
1108    CHUNK = 64
1109    chunk_indices = prepare_chunk_indices(cu_seqlens, CHUNK)
1110    chunk_offsets = prepare_chunk_offsets(cu_seqlens, CHUNK)
1111    chunk_offsets_ini = prepare_chunk_offsets_with_ini(cu_seqlens, CHUNK)
1112    NS, NC = len(cu_seqlens) - 1, chunk_indices.shape[0]
1113    NC_WITH_INI = NC + NS
1114    out = torch.empty_like(v)
1115    use_aot = os.environ.get('FORGE_USE_AOT', '0')
1116    FORGE_USE_AOT = True if use_aot.lower() in ['1', 'true', 'yes'] else
         None
1117
1118    def alloc_fn(size: int, alignment: int, stream: int):
1119        return torch.empty(size, device='cuda', dtype=torch.int8)
1120
1121    triton.set_allocator(alloc_fn)
1122
1123    def grid(meta):
1124        BLK_K = meta['BLK_K']
1125        BLK_V = meta['BLK_V']
1126        NUM_BLK_K = (K + BLK_K - 1) // BLK_K
1127        NUM_BLK_V = (V + BLK_V - 1) // BLK_V
1128        NUM_BLK_KV = NUM_BLK_K * NUM_BLK_V
1129        H_MUL_NS = H * NS
1130        return (NUM_BLK_KV, H, NC,)
1131
1132    if FORGE_USE_AOT is None:
1133        merge_mode_kernel[grid](
1134            chunk_decay=chunk_decay,
1135            stride_d0_nc_with_ini_chunk_decay=H,
1136            v=v,
1137            k=k,
1138            q=q,
1139            g=g,
1140            prev_rank_state_sum=prev_rank_state_sum,
1141            stride_d0_ns_prev_rank_state_sum=H*K*V,
1142            chunk_state=chunk_state,
1143            scale=scale,
1144            cu_seqlens=cu_seqlens,
1145            H=H,
1146            NS=NS,
1147            K=K,
1148            V=V,
1149            CHUNK=CHUNK,
1150            H_MUL_NS=H * NS,
1151            out_0=out,
1152            chunk_indices=chunk_indices,
1153            NC=NC,
1154            FUSE_SP_STATE_UPDATE=fuse_sp_update,
1155        )
1156
1157    else:
1158
1159        from forge.aot_utils import forge_aot_ops
1160
1161        algo_info = forge_aot_ops.merge_mode_kernel__triton_algo_info_t()
1162        for _k, _v in get_merge_mode_kernel_info(NS, T, H, K, V).items():
1163            setattr(algo_info, _k, _v)
1164        forge_aot_ops.merge_mode_kernel(
1165            0,  # torch.cuda.current_stream().cuda_stream,
1166            q.data_ptr(),  # q
```

```
1167            k.data_ptr(),   # k
1168            v.data_ptr(),   # v
1169            g.data_ptr(),   # g
1170            chunk_state.data_ptr(),   # chunk_state
1171            scale,   # scale
1172            prev_rank_state_sum.data_ptr() if prev_rank_state_sum else 0,
         # prev_rank_state_sum
1173            H*K*V,   # stride_d0_ns_prev_rank_state_sum
1174            chunk_decay.data_ptr() if chunk_decay else 0,   # chunk_decay
1175            H,   # stride_d0_nc_with_ini_chunk_decay
1176            out.data_ptr(),   # out_0
1177            cu_seqlens.data_ptr(),   # cu_seqlens
1178            NS,   # NS
1179            H * NS,   # H_MUL_NS
1180            chunk_indices.data_ptr(),   # chunk_indices
1181            NC,   # NC
1182            algo_info,
1183        )
1184
1185    cached_results.update({})
1186    return out
1187
1188 def fused_op():
1189    cached_results = {}
1190    # AUTO: precompute decay outside
1191    g_cumsum = chunk_local_cumsum(g[None, ...], 64, cu_seqlens=cu_seqlens
       ).squeeze(0)
1192    updated_states = launch_fuse_chunk_decay(
1193        g=g_cumsum,
1194        prev_s=prev_s,
1195        v=v,
1196        k=k,
1197        cu_seqlens=cu_seqlens,
1198        cached_results=cached_results,
1199        dist_scan=dist_scan,
1200        lazy_update=lazy_update,
1201    )
1202    chunk_state = updated_states['chunk_state']
1203    prev_rank_state_sum = updated_states.get('prev_rank_state_sum', None)
1204    chunk_decay = updated_states.get('chunk_decay', None)
1205    o = launch_merge_mode(
1206        chunk_decay=chunk_decay,
1207        v=v,
1208        k=k,
1209        q=q,
1210        g=g_cumsum,
1211        prev_rank_state_sum=prev_rank_state_sum,
1212        chunk_state=chunk_state,
1213        scale=scale,
1214        cu_seqlens=cu_seqlens,
1215        cached_results=cached_results,
1216        fuse_sp_update=fuse_sp_update,
1217    )
1218    return o, chunk_state
```

Listing 2: Generated Scalar GLA kernel by Forge

## G.2 IMPLEMENTATION FOR DELTANET

```
1 def chunk_mode_deltanet(k: Tensor, v: Tensor, b: Tensor) -> Tensor:
2    I = forge.identity(k, k.size(0))
3    # Note: forge handles the specific fp32 accumulation
4    T = (I + forge.matmul_out_fp32(k * b[..., None], k.T).tril(-1)).
      inverse().to(k.dtype)
```

```
 5      U = T @ (v * b[..., None])
 6      W = T @ (k * b[..., None])
 7      # Explicitly cache results for reuse in other phases
 8      forge.cache_result(U, "u")
 9      forge.cache_result(W, "w")
10      S = k.T @ U
11      return S
12
13  def decay_mode_deltanet(prev_s: Tensor, k: Tensor, w: Tensor, chunk_state
        : Tensor) -> Tensor:
14      """
15      Inter-Chunk State Propagation
16      """
17      # Calculate decay matrix based on cached W
18      decay = forge.identity(k, k.size(1)) - k.T @ w
19      return decay @ prev_s.to(decay.dtype) + chunk_state
20
21  def merge_mode_deltanet(q: Tensor, k: Tensor, u: Tensor, w: Tensor,
        chunk_state: Tensor, scale: Tensor) -> Tensor:
22      """
23      Output Merging
24      Note: 'u' and 'w' are automatically injected from the cached results
25      """
26      new_v = u - w @ chunk_state
27      return ((q @ k.T).tril(0) @ new_v + q @ chunk_state) * scale
```

Listing 3: Forge Implementation of DeltaNet with Intermediate Result Caching

