# OpenReview forum: "FlexLinearAttention: Compiling a Unified Abstraction into Scalable Kernels for Linear Attention"
_ICLR.cc/2026/Conference — ICLR 2026 Poster_

### Official Review · Reviewer_r3XC · 2025-10-21

**Soundness:** 3
**Presentation:** 3
**Contribution:** 3
**Rating:** 6
**Confidence:** 4

**Summary:**

This paper presents Forge, a domain-specific compiler designed specifically for linear attention mechanisms. Forge abstracts the computation into three canonical stages based on chunk-wise parallelization: intra-chunk computation, inter-chunk state propagation, and output merging. This unified abstraction both simplifies the development of new linear attention variants (such as Mamba, RetNet, RWKV, GLA, HGRN, and GDN) and enables systematic optimizations for both single-device and distributed scenarios. Forge leverages Triton-Distributed as its backend to generate high-performance kernels that fuse computation and communication, and incorporates an adaptive scheduling system for parallelism and a static dispatcher to reduce runtime overhead. Experimental results demonstrate that Forge matches or outperforms state-of-the-art hand-written kernels (e.g., Flash Linear Attention) in both performance and scalability, while also requiring much less manual effort to implement new variants.

**Strengths:**

1. Proposes a unified, three-phase abstraction for linear attention that captures a wide range of state-of-the-art variants, enabling easy mapping from high-level algorithms to efficient implementations.
2. Automates the generation of high-performance, hardware-aware kernels from simple PyTorch code, reducing the barrier for researchers to experiment with or deploy new linear attention models.
3. Integrates native support for fine-grained, tile-level compute-communication fusion using Triton-Distributed, improving distributed execution and network bandwidth utilization compared to existing methods.
4. Demonstrates strong empirical results, with Forge-generated kernels achieving up to 4.9x speedup over expert-tuned baselines and near-linear weak scaling up to 128 GPUs and 16 million tokens.
5. Includes practical system-level optimizations such as ahead-of-time compilation and static dispatching to eliminate runtime overheads, particularly beneficial for short to medium sequence lengths.
6. Provides evidence that the abstraction is expressive enough to cover a broad family of linear attention models with minimal code changes.

**Weaknesses:**

1. As a compiler-based framework, the proposed abstraction is tailored specifically for linear attention and lacks extensibility to other attention mechanisms, which limits its general applicability.
2. Although Forge demonstrates competitive performance compared to Flash Linear Attention, the paper does not sufficiently emphasize its advantages, particularly in terms of implementation flexibility and extensibility relative to Triton-based Flash Linear Attention.
3. While Forge automates optimization, the actual benefits in terms of code brevity or maintainability versus existing Triton-based manual approaches are not quantified in user studies or qualitative analysis.
4. The distributed scaling results may be somewhat overstated: for example, in Figure 1, the single-GPU latency is 9.2ms, while 4 GPUs achieve 2.7ms. The real scaling efficiency is about 85%, not precisely "near-linear" as asserted.
5. The distributed experiments (Section 4.2) use H20 GPUs (with slower communication) and BatchSize=4, whereas single-GPU experiments use H100 GPUs and BatchSize=1. This difference in hardware and batch size may exaggerate the benefits of compute-communication overlap and makes the experimental comparison less fair.

**Questions:**

1. According to the experimental results, Flash Linear Attention already supports multiple linear attention variants and delivers strong performance across them. While Forge’s three-phase abstraction enables users to easily implement different variants, Flash Linear Attention is also Triton-based, which inherently provides customization capabilities. How does Forge compare to Flash Linear Attention in terms of flexibility when supporting diverse linear attention variants?
2. How does the Parallelism Scheduler mentioned in Appendix B adapt to distributed environments? Are there differences in scheduling strategies between single-GPU, intra-node, and inter-node scenarios?
3. Are there any limitations or edge cases where the three-phase abstraction might restrict the expression of certain linear attention updates, particularly for future architectures that may involve more complex dependencies?
4. What is the overhead (if any) of using Forge's compilation pipeline compared to maintaining optimized hand-written kernels, especially as new hardware or Triton versions are released?

---

> ### Author Response · Authors · 2025-11-23
> **Reply part 1**
>
> We sincerely thank you for your invaluable feedback. Your deep insights into system-level design, particularly regarding the distributed adaptation of our scheduler and the nuances of experimental fairness, have prompted us to significantly strengthen our evaluation and articulate our system architecture with greater precision. Below we try our best to address your concerns:
>
> > ### W1: tailored specifically for linear attention and lacks extensibility to other attention mechanisms.
>
> In fact, this is a very fundamental issue. We want to emphasize that Forge's ability to so flexibly express various linear attention variants and achieve distributed scaling is precisely because it focuses on it. If we increase generalization, we pay for it. The inevitable result is that we have to provide a less intuitive abstraction.
>
> To answer this question, we add an addtional section in **appendix E** to discuss about how to extent Forge to softmax attention. Here is a brief rebuttal to reviewer's argument that focusing on linear attention is a drawback:
>
> expanding Forge to support Softmax attention is theoretically feasible and would require two primary extensions to our current abstraction:
> 1. **Generalizing the "State" Definition:Current (Linear)**: In Linear Attention, the "Inter-Chunk State" is a feature map matrix (e.g., $S \in \mathbb{R}^{d \times d}$) that follows a linear recurrence. **Required (Softmax)**: For Softmax attention, the "associative state" that needs to be propagated between chunks consists of normalization statistics (the running maximum $m$ and the running sum $l$) rather than a generative hidden state. Expanding Forge would require allowing the "Inter-Chunk" phase to carry these scalar/vector statistics.
> 2. **Introducing a "Rescaling" Primitive**:Current (Linear): The "Merge" phase in Forge is typically a linear combination at chunk-level. **Required (Softmax)**: The "Merge" phase in Softmax attention requires rescaling. When merging partial results from two blocks, the output of the first block must be scaled down based on the difference between its local maximum and the new global maximum ($O_{new} = O_1 \times e^{m_1 - m_{new}} + \dots$). Adding a rescale(output, old_stats, new_stats) primitive to the Forge DSL would be the key technical step to supporting Softmax.
>
> **Why we may not do that**: While technically possible, **we deliberately chose to specialize in Linear Attention to maximize expressiveness**. History suggests that trying to make a DSL "universal" often compromises its ability to handle complex, domain-specific patterns.
>
> 1. **Evidence**: A recent work AttentionEngine [1], attempts to support both softmax and linear attention. However, to maintain this generality, its abstraction is too rigid to express advanced linear attention variants like the Delta Rule or Vector-Gated GLA—both of which Forge supports effortlessly.
> 2. **Ecosystem**: The Softmax domain is already well-served by excellent works like FlexAttention (also, a compiler-driven framework) and FlashAttention-2/3/4. In contrast, Linear Attention researchers are inventing new variants monthly, each requiring a bespoke kernel. Forge solves this specific "N-to-1" compiler challenge, which is currently a larger bottleneck for the community than optimizing Softmax attention. **We see greater value in solving the "fragmentation problem" for Linear Attention.**
>
> **Transferable Insights**: Finally, while our abstraction is specialized, the system-level innovations we built for Forge are generalizable. Our AOT compilation pipeline and static dispatcher (which eliminate runtime overheads) are universal optimizations that could be directly adopted by Softmax-focused DSLs (like FlexAttention) to improve their performance on short sequences.
>
> [1] Chen, Feiyang, et al. "Attentionengine: A versatile framework for efficient attention mechanisms on diverse hardware platforms." arXiv preprint arXiv:2502.15349 (2025).

---

> ### Author Response · Authors · 2025-11-23
> **Reply part 2**
>
> > ### W2 + Q1: Implementation flexibility and extensibility relative to Triton-based Flash Linear Attention.
>
> We appreciate you raising this fundamental question regarding the "Library vs. Compiler" trade-off. We agree that writing raw kernels (as FLA does) offers the ultimate low-level control. However, we argue that "control" does not equal "practical flexibility" for the vast majority of researchers. We clarify the distinct advantages of Forge below:
>
> 1. While FLA is indeed Triton-based and customizable, doing so effectively imposes a high "expertise tax". To implement a new variant in FLA, a user cannot simply update a formula. They must often rewrite the kernel's inner loop, manually manage synchronization, tiling strategy, and resolve potential shared memory capacity problems. **This makes "flexibility" accessible only to kernel experts (as the maintainer of FLA, but not the majority of algorithm researchers). In contrast, Forge abstracts away these hardware-specific complexities.** This dramatically lowers the barrier to entry, transforming what is currently a complex engineering challenge into a simple mathematical definition problem. This velocity of iteration is Forge's primary advantage.
> 2. As Flexibility is difficult to quantify directly, we try our best to demonstrate it through the diversity of algorithms Forge supports in paper. With our abstraction design, Forge covers variants ranging from simple decays (RetNet) to complex data-dependent updates (DeltaNet) and vector-gated mechanisms. We deliberately constrained Forge to the Linear Attention domain, which allows our abstraction to remain expressive enough to capture complex patterns (like the Delta Rule) that generic frameworks often miss, while still being significantly more programmable than raw Triton.
>
> If "flexibility" is defined as the ability to manually optimize a specific instruction, FLA/Hand-Written Kernel wins. But if "flexibility" is defined as the velocity with which a researcher can implement and test a novel mathematical idea with high performance, Forge provides a decisive advantage.
>
> > ### W3: Qualitative analysis of code brevity or maintainability.
>
> Thank you for pointing this out. We have already incorporated the qualitative analysis into our manuscript (an additional section in **Appendix G**). Using Forge to implement a scalar GLA kernel only involves **50+** lines of code (LoC), which the generated triton code and host launcher is around **1200 LoC**. Even without including the static dispatcher (our AOT optimization) and host launcher in count, the generated Triton kernel alone has over **400 LoC**. And this is only partially generated content, because Forge also generates code for other parallel strategies.
>
> As for maintainability, we illustrate this with an example: In Forge, generating TMA instructions only requires one additional compilation pass (200+ LoC), while the corresponding handwritten kernel requires modifications to each kernel.
>
> > ### W4: The distributed scaling results may be somewhat overstated.
>
> We are very grateful to the reviewer for pointing this out, and we have revised this statement in our latest manuscript to "scale with near-linear efficiency on scalar gated linear attention to 16 million tokens on 128 GPUs" (Line 026).
>
> Our claim of **near-linear** scaling in a distributed environment is based on our weak scaling experiments in Section 4.2: keeping the workload per device constant, scaling up the number of GPUs resulted in almost no increase in overall latency: e.g., **scaling scalar GLA from 4 to 128 GPUs with a 16K sequence length per device only increased overall latency by 8%**. However, we must acknowledge that scaling is difficult to achieve ideally when the workload per device is not enough or when state updates are more complex. **We have honestly demonstrated these results, such as the GDN results in Section 4.2, and Figure 1 as pointed out by the reviewer.**
>
> > ### W5: The Experiments setting makes the experimental comparison less fair.
>
> We appreciate you checking the experimental setup carefully. **The reason why we use H20 GPUs for distributed experiments is we only had access to a single H100 node (which we used for single-device test) and a cluster with 128 H20.**
> 1. We emphasize that the comparison remains strictly fair because both the baseline (LASP2) and Forge were evaluated under the exact same hardware and batch size configurations on the H20 cluster.
> 2. The H20 cluster has the same bandwidth as the H100 cluster but a lower computation throughput (148 v.s. 989.5 TFLOPS). This actually creates an environment where communication is less stressful.
>
> To address your concern about the hardware gap, we have conducted additional experiments on H100 GPUs. (using our available 4-GPU and 8-GPU intra-node configurations). We evaluated on **batch=1 (instead of batchsize=4)**, other settings remain the same with origin in paper. The results is included in **Appendix F**.

---

> ### Author Response · Authors · 2025-11-23
> **Reply part 3**
>
> > ### Q2: How does the Parallelism Scheduler mentioned in Appendix B adapt to distributed environments? Are there differences in scheduling strategies between single-GPU, intra-node, and inter-node scenarios?
>
> Thank you for this detailed system-level question. We would like to clarify the scope of the Parallelism Scheduler and our current distributed strategy.
> 1. **Scope of the Parallelism Scheduler**: The Parallelism Scheduler described in Appendix B is designed primarily for single-GPU optimization. Its role is to automatically select the optimal tiling strategy and grid configuration to maximize local hardware utilization given the input shape.
>     - **In Distributed Settings**: The distributed execution is composed of local computations on each rank followed by state communication. The Parallelism Scheduler optimizes how the "local partial state" is computed on each GPU. The communication of the final state between ranks is handled by a separate distributed runtime layer, which operates orthogonally to the local instruction scheduling.
>
> 2. **Intra-node vs. Inter-node Strategies**
>     - **Current Strategy (Unified)**: Currently, Forge employs a unified distributed strategy for both intra-node and inter-node scenarios. We use a fine-grained pipelined send/recv mechanism that is fused with the computation at tile level. This allows us to hide the communication latency of the state transfer regardless of the physical topology.
>     - **Future Optimization (Differentiated)**: We fully agree with your insight on differentiating between intra-node and inter-node communication: We are actively exploring a specialized intra-node optimization that utilizes the high-bandwidth NVLS domain. Instead of pipelined P2P passing, we can employ customized "one-shot" collective communication (aggregating all local states via a specialized low-latency kernel, with communication volume roughly half that of a standard `all-gather`).
>
> **We appreciate you pointing this out, as it confirms our roadmap for future system-level enhancements. We are open to further discussion if you have specific thoughts!**
>
> > ### Q3: Can Forge adapt to future change as model architecture evolves.
>
> We thank reviewers for raising this foundational question. To address this concern, we have added a discussion section to the paper (**appendix D**) to explain why we believe that Forge's abstraction design and abstraction levels are flexible enough to adapt to the future.
>
> **Algorithmic Expressiveness: Grounded in Associativity**: The "future-proof" nature of Forge's abstraction (Intra, Inter, Merge) comes from the fact that it is not a heuristic, but a direct mapping of the associative property inherent to efficient sequence modeling.
>
> - **Why it works**: Almost all "Linear Attention" or "SSD" models aim to achieve $O(N)$ complexity by formulating the attention mechanism as a recurrence (RNN-like) or a parallel prefix scan. Mathematically, any algorithm that can be formulated as chunk parallel form fits our abstraction.
> - **Handling Complex Dependencies**: Forge is expressive enough to handle complex state updates found in modern variants. For example, the Delta Rule (which involves a data-dependent update) and element-wise decay are fully supported because they preserve the chunk-wise associative structure.
> - **The Limitation (Edge Cases)**: The limitation of our abstraction lies strictly in non-associative recurrences. If a future architecture introduces a dependency where the state update $h_t = f(h_{t-1}, x_t)$ involves a non-linear function $f$ that prevents parallel scan (e.g., passing the hidden state through a complex MLP at every step before the next update), Forge would not apply. However, such models essentially abandon the parallel advantage, making them unlikely candidates for the high-efficiency regime Forge targets. For example, TTT[1] can be viewed as linear attention to some extent (from a fast-weight perspective), but its updates are non-linear, so we cannot express it within Forge. However, it's worth noting that this also means that TTT cannot be as efficient as previous linear attention methods.
>
> [1] Sun, Yu, et al. "Learning to (learn at test time): Rnns with expressive hidden states." arXiv preprint arXiv:2407.04620 (2024).

---

> ### Author Response · Authors · 2025-11-23
> **Reply part 4**
>
> > #### Q4: What is the overhead (if any) of using Forge's compilation pipeline compared to maintaining optimized handwritten kernels, especially as new hardware or Triton versions are released?
>
> We thank you for this insightful question regarding the long-term engineering trade-offs of compiler-based systems. We analyze the "overhead" from two perspectives: Performance and Maintenance Effort.
> 1. **Performance Overhead**: Negligible to Negative: Unlike JIT approaches, Forge incurs **zero runtime overhead** during execution. The compilation happens Ahead-of-Time (AOT). The generated artifacts are native Triton kernels (essentially cubin module) that run directly on the GPU. AOT takes some time to configure various shape searches. **Fortunately, this process is a one-time event, and this overhead is consistent with traditional AOT systems (such as hand-written CUDA kernels).**
> 2. **Maintenance Overhead: Reduced (O(1) vs O(N))**. The primary advantage of Forge is the reduction of maintenance burden as hardware and software stacks evolve.
>     - **Hand-Written Kernels (The O(N) Trap)**: In a library of hand-written kernels (e.g. FLA), supporting a new hardware feature (like Hopper's TMA) requires manually rewriting and verifying the kernel for every supported variant. This leads to a linear maintenance cost O(N) where N is the number of variants.
>     - **Forge (The O(1) Solution)**: With Forge, the hardware-specific optimizations are encapsulated in the Compiler Backend. Adapting to a new GPU architecture or a major Triton version update requires modifying only the backend code generation logic once. Once updated, all user-defined models automatically inherit the new hardware capabilities without a single line of change in the model code.
>     - **Dependency Management**: Forge sits atop the Triton compiler. As Triton evolves to support new hardware (e.g., Blackwell), Forge leverages these upstream improvements automatically. **Furthermore, since Triton's API is very stable, we only need to incur the corresponding adaptation costs for specific new feature APIs.**
>
> We also include an additional discussion about hardware sustainability of Forge in **Appendix D.2.**
>
> ---
>
> We truly enjoyed this constructive dialogue from a system software perspective and are excited to find your suggestions (especially regarding potential intra/inter node optimization)  coincides with our own thoughts. We hope our new H100 experiments and detailed clarifications have fully addressed your concerns, and we would be delighted to discuss any further questions you might have!

---

> > ### Comment · Reviewer_r3XC · 2025-11-25
> >
> > Thank you very much for the authors' response and for accepting our suggestions. The relevant revisions and experiments have also been presented in the revised manuscript. I will maintain my original score and leave it to the Area Chair to make a comprehensive decision regarding the acceptance of the paper.

---

### Official Review · Reviewer_RnXQ · 2025-10-26

**Soundness:** 3
**Presentation:** 4
**Contribution:** 3
**Rating:** 6
**Confidence:** 3

**Summary:**

Forge is a DSL + compiler for linear attention. It abstracts linear attention variants to three definitions, and uses a compilation backend with a distributed version of triton-lang to optimize linear attention across computation and communication resources. Forge showcases its efficiency by comparing against Flash Linear Attention, a library of hand-designed kernels for linear attention, and performs on par or better than it.

**Strengths:**

Thank you for submitting your work. This was a pleasant read and I am optimistic it will be valuable to the community. I found the presentation to be very well-done, and the explanations of how the DSL is translated to intra and inter-GPU parallelized code was interesting to read. The experiments are expansive and the results are promising.

**Weaknesses:**

There are two issues I am concerned about.
* The central issue of abstractions is being future-proof. This boils down to 1) whether this abstraction is expressive enough for a wide range of designs, and 2) whether maintaining the abstraction (translation and compilation) is sustainable with hardware changes. These questions are challenging to answer, since to some extent they involve making educated guesses. But I still would have liked to see some discussion about these points in a paper of this sort.
* I could not find an anonymized code repository in the text. It is odd not to share code for papers with DSL contributions. Please add an anonymized code repository link to the paper text.

**Questions:**

* Why did the authors find this particular abstraction more expressive than the rest?
* Are there linear attention variants this abstraction does not support? Is there an example of this from recent publications in the community?
* Why did the authors not include source code with the text?
* Can you include the chunk, decay and merge sections for the five variants studied in the paper? This would be ideally be in a code repository, but in absence of that it should be added to the appendix.
* Have the authors read AttentionEngine [1]? Seems like relevant work.

[1] Chen, Feiyang, et al. "Attentionengine: A versatile framework for efficient attention mechanisms on diverse hardware platforms." arXiv preprint arXiv:2502.15349 (2025).

---

> ### Author Response · Authors · 2025-11-23
> **Reply part 1**
>
> We sincerely appreciate reviewer's 'excellent' rating on our presentation and your forward-looking questions regarding the sustainability of our abstraction.  We hope our response could address your concerns!
>
> > ### W1: Can Forge adapt to future change in both algorithm and hardware perspective. Why we claim this abstraction more expressive than the rest? Are there any variants that Forge can not express?
>
> We thank reviewers for raising this foundational question. To address this issue, we have added a discussion section to the manualscript (**appendix D**) to explain why we believe that Forge's current abstraction design and abstraction levels are flexible enough to adapt to the future.
>
> 1. **Algorithmic Expressiveness: Grounded in Associativity (why we claim this abstraction work)**: The "future-proof" nature of Forge's abstraction (Intra, Inter, Merge) comes from the fact that it is not a heuristic, but a direct mapping of the associative property inherent to efficient sequence modeling.
>     - **Why it works**: Almost all "Linear Attention" or "SSD" models aim to achieve $O(N)$ complexity by formulating the attention mechanism as a linear recurrence (RNN-like) or a parallel prefix scan. Mathematically, any algorithm that can be formulated to chunk-level parallel form fits our abstraction.
>     - **Handling Complex Dependencies**: Forge is expressive enough to handle complex state updates found in modern variants. For example, the Delta Rule (which involves a data-dependent update) and element-wise decay are fully supported because they preserve the chunk-wise associative structure.
>     - **The Limitation (Edge Cases)**: The limitation of our abstraction lies strictly in non-associative recurrences. If a future architecture introduces a dependency where the state update $h_t = f( h_{t-1}, x_t)$ involves a non-linear function $f$ that prevents parallel scan (e.g., passing the hidden state through a complex MLP at every step before the next update), Forge would not apply. However, such models essentially abandon the parallel advantage, making them unlikely candidates for the high-efficiency regime Forge targets. For example, TTT [1] can be viewed as linear attention to some extent (from a fast-weight perspective), **but its updates are non-linear, so we cannot express it within Forge**. However, it's worth noting that this also means that TTT cannot be as efficient as previous linear attention methods.
>
> 2. **Hardware Sustainability: Decoupling via Intermediate DSLs**: Our approach to sustainability relies on a strict hierarchical decoupling between the algorithm and the hardware instructions.
>     - **Layered Compilation**: Forge does not directly emit hardware-specific assembly. **Instead, it functions as a high-level compiler that lowers the algorithm into an intermediate kernel DSL**. In our current implementation, we target Triton, effectively delegating low-level complexities such as register allocation, instruction scheduling, and tensor core management to the Triton compiler.
>     - **Backend Agnostic**: This design choice makes Forge inherently adaptable. While we currently use Triton, architecture allows us to retarget the mapping layer to other emerging DSLs (e.g., CuTeDSL or TileLang) if they offer better performance or hardware support in the future. We only sparingly embed inline PTX for specific extensions not yet supported by the backend.
>     - **Sustainability**: Consequently, when new hardware architectures (e.g., Blackwell, Rubin) arrive, we leverage the updates made by the community to these intermediate compilers. We do not need to rewrite the entire system; we simply update the mapping strategy. Kernel DSLs like Triton typically have relative stable API, so the update of compilation flow is not expected to be too complicated.
>
> [1] Sun, Yu, et al. "Learning to (learn at test time): Rnns with expressive hidden states." arXiv preprint arXiv:2407.04620 (2024)

---

> ### Author Response · Authors · 2025-11-23
> **Reply part 2**
>
> > ### W2: Code example and open source repo.
>
> We thank you for emphasizing the importance of reproducibility and for requesting the implementation details.
>
> 1. Code Examples in **Appendix G**: Per your request, we have added the source code for the chunk, decay, and merge mode of some variants in Appendix G of the revised manualscript.
>     - **Note**: As the full kernels generated by Forge can be quite verbose, we have focused on providing the core DSL specifications and one generated example (around 1200 lines of code). **We are happy to provide additional code segments during the discussion period if there are specific implementation details you wish to examine further.**
>
> 2. Repository and Open Source Plan We sincerely apologize for not providing an anonymous repository link at this stage. Due to strict institutional review and compliance policies within our organization, we are currently unable to release the full codebase externally during the review process.
>     - Commitment: we guarantee that the full project will be open-sourced upon publication (or when we put this work on arXiv).
>
> Furthermore, to maximize the practical impact of our work, we are actively working on integrating Forge's backend directly into the torch.compile (actually, Inductor) stack. Our goal is to make these optimizations available natively to the broader PyTorch ecosystem, moving beyond a standalone research prototype.
>
> > ###  W3: Related work AttentionEngine.
>
> Thank you for bringing this work to our attention. We have reviewed AttentionEngine and included a detailed discussion about the trade-off between "unified frameworks" and "specialized compilers" in our revised manualscript (**Appendix E**).
>
> We believe a comparison with AttentionEngine actually highlights the core value proposition of Forge:
>
>    - **Generality vs. Expressiveness**: AttentionEngine aims to provide a versatile framework covering both Softmax and Linear Attention. However, our analysis suggests that this broad scope necessitates a rigid abstraction that limits its ability to express complex Linear Attention patterns.
>    - **Specific Limitations**: For instance, the abstraction in AttentionEngine struggles to support advanced variants like the Delta Rule (which requires data-dependent updates) or Vector-Gated GLA.
>    - **Forge's Advantage**: In contrast, by specializing strictly in Linear Attention, Forge’s abstraction handles these complex dependencies natively while enabling deep, domain-specific optimizations (like our specialized tiling for recurrence) that a general-purpose engine might miss.
>
> And we checked out the code of AttentionEngine. It depends on predefined tilelang template specializations for code generation. **We also added a performance comparison with tilelang (see our reply to Reviewer `VUej` (part 1) if interested)**.
>
> ---
>
> Your feedback on future-proofing and the comparison with concurrent works like AttentionEngine has added significant depth to our discussion on expressiveness. Furthermore, your request for code examples has directly improved the transparency and reproducibility of our submission. We are committed to the open-source roadmap we described and hope that our revisions and detailed responses confirm the long-term value of Forge. **We'd love to provide additional code snippets at any time, and we hope these answers address your concerns!**

---

### Official Review · Reviewer_PFK1 · 2025-10-28

**Soundness:** 3
**Presentation:** 3
**Contribution:** 2
**Rating:** 4
**Confidence:** 2

**Summary:**

What about the following summary, am I missing anything?
- Forge introduces a domain-specific language (DSL) for linear attention kernels - this is the key novelty and the abstractions defined there
- The DSL is compiled to triton kernels. The compiler can do optimizations such as compute communication fusion, and a range of targeted optimization of system bottlenecks
- in single GPU, forge generated kernels can give an increase of 1.01-4.9x speedup compared to hand-tuned kernels
- Forge can also generate distributed kernels accross multiple GPUs, which is demonstrated up to 128

Overall, forge is a well thought through engineering solution that can help to develop linear attention kernels with good performance, without the effort of manually implementation

**Strengths:**

- well-engineered compiler system that automates generation of linear attention kernels, both for single GPU and distributed kernels
- integrates well into existing pipelines such as Triton, this will make it easier to adopt.
- important optimizations for compute- communication fusion, and ahead-of-time (AOT) compilation
- good empirical results, showing 1.01x-4.9x speedup over FlashLinearAttention on single GPUs and near-linear scaling up to 128 GPUs
- practical significance for research on models that leverage new linear attention mechanism, reducing manual kernel engineering effort which is one of the limitations to scale more researchy models

**Weaknesses:**

- limited novelty (the DSL/abstraction for linear attention), but good integration of existing work
- too narrow. linear attention is not widely used, and mainly for more researchy models. Not relevant for predominant model architectures. This limits the practical use of this framework to a very specific set of research explorations. Expanding this to more common forms of attention would be desirable.

**Questions:**

I would love to understand what would be required to expand this framework to softmax based attention. The authors discuss associativity at  the beginning of the paper, but associativity also has been exploited for optimizations with softmax attention (see e.g. LeanAttention)

---

> ### Author Response · Authors · 2025-11-23
> **Reply part 1**
>
> Thanks for your recognition of Forge's engineering efforts and practical significance. Your discussion regarding softmax attention has prompted us to give this project a lot of thought, and we are attempting to address your concerns and sharing our thoughts here.
>
> > ### W1: Concerns regarding "Limited Novelty" (DSL/abstraction) and "Integration of existing work".
>
> We respectfully disagree that Forge is merely an integration of existing tools. While we leverage existing backends (like Triton), our core contribution lies in defining the missing system architecture that bridges the gap between high-level Linear Attention math and low-level hardware efficiency, a gap that standard integration fails to close.
>
> 1. The Abstraction is the Innovation (we use "MapReduce" as an Analogy):
>
>     While the mathematical formulation of Linear Attention exists, identifying the minimal unified abstraction (Intra-Chunk, Inter-Chunk, Merge) that simultaneously covers diverse variants (RetNet, Mamba, GLA, RWKV) and maps efficiently to hardware intrinsic is a non-trivial research contribution.
>
>    **Why this is novel**: Similar to how MapReduce did not invent "map" or "reduce" functions but innovated by abstracting them into a scalable distributed system, Forge abstracts the complex recurrence of Linear Attention into a unified compilation target. This abstraction transforms an $O(N \times M)$ engineering burden ($N$ models on $M$ hardware) into an $O(1)$ compiler problem, which is a novel conceptual leap in this domain.
>
> 2. Novelty in Distributed System Design (Beyond Engineering Integration)
>
>     To the best of our knowledge, Forge is the first work to automate Computation-Communication Fusion for Linear Attention Context Parallelism. This is fundamentally different from "integrating" a communication library (like NCCL) with a compiler (like `torch.compile`):
>     - **Failure of Naive Engineering Integration**: As shown in our ablation studies, simply combining NCCL leads to severe latency because standard collectives cannot penetrate the tight recurrent loop of linear attention.
>     - **Forge's Novel Design**: Our compiler automatically analyzes the data dependency of the "Inter-Chunk" phase and injects fine-grained, pipelined P2P communication instructions inside the computation kernel. This design requires a deep understanding of both distributed systems and the algebraic structure of linear attention, something that cannot be achieved by simply "using" existing tools.
>
> Overall, Forge provides the first systematic solution to the "fragmentation" and "scalability" problems in Linear Attention. The novelty lies not in the components used, but in the compiler architecture that enables these components to work together with mathematical correctness and hardware optimality, achieving results that manual engineering or naive integration cannot replicate.

---

> ### Author Response · Authors · 2025-11-23
> **Reply part 2**
>
> > ### W2: too narrow. linear attention is not widely used, and mainly for more research models.
>
> We acknowledge that softmax attention remains the predominant architecture in current mainstream Transformer models. However, we respectfully disagree with the reviewer’s assertion that "linear attention is not widely used" and therefore lacks practical relevance. In fact, there is rapidly growing adoption of linear-attention and hybrid-attention architectures across both industry and academia.
> 1. **Model-side adoption.**
>  Large-scale models built on linear attention have already been deployed in real-world products. For example, Qwen-Next (80B) is commercially served on the Qwen platform and achieves state-of-the-art results on several third-party benchmarks (e.g., surpassing DeepSeek-R1 on AMO-Bench) [2]. NVIDIA has released a production T2I model based on linear attention [3]. MiniMax has scaled linear-attention models up to 465B parameters [4], and most recently, Moonshot introduced the 48B KDA model employing a linear-attention architecture [5]. These examples demonstrate that linear attention is no longer limited to “researchy” prototypes but is actively being scaled, deployed, and commercialized.
> 2. **Infrastructure-side adoption.**
>  Modern inference frameworks are also embracing linear attention as a first-class citizen. Both *vLLM* and *sglang* have recently incorporated explicit support and memory-management optimizations tailored for linear-attention models [1,6]. Importantly, linear attention introduces fundamentally different computation and communication patterns compared to softmax attention; thus, dedicated abstractions and code-generation systems are required. Forge is designed precisely to address these emerging needs.
> Overall, the evidence suggests that linear attention is becoming increasingly relevant for both large-scale model development and practical deployment.Forge provides timely and necessary systems support for this evolving class of architecture.
>
> [1] Zhang, Chen, et al. "JENGA: Effective memory management for serving LLM with heterogeneity." Proceedings of the ACM SIGOPS 31st Symposium on Operating Systems Principles. 2025.
>
> [2] An, Shengnan, et al. "AMO-Bench: Large Language Models Still Struggle in High School Math Competitions." arXiv preprint arXiv:2510.26768 (2025).
>
> [3] Xie, Enze, et al. "Sana: Efficient high-resolution image synthesis with linear diffusion transformers." arXiv preprint arXiv:2410.10629 (2024).
>
> [4] Li, Aonian, et al. "Minimax-01: Scaling foundation models with lightning attention." arXiv preprint arXiv:2501.08313 (2025).
>
> [5] Team, Kimi, et al. "Kimi Linear: An Expressive, Efficient Attention Architecture." arXiv preprint arXiv:2510.26692 (2025).
>
> [6] Pan, Rui, et al. "Marconi: Prefix caching for the era of hybrid LLMs." arXiv preprint arXiv:2411.19379 (2024).

---

> ### Author Response · Authors · 2025-11-23
> **Reply part 3**
>
> > ### Q1: Can forge be expanded to softmax attention? If so,  what would be required (e.g. associativity)?
>
> This is an insightful question! You are correct that Softmax attention also possesses associative properties (via the online softmax trick), which have been exploited by optimizations like FlashAttention and LeanAttention. To answer your question directly: expanding Forge to support Softmax attention is theoretically feasible and would require two primary extensions to our current abstraction. To address this concern, we have added a discussion section to the paper (**appendix D**) to discuss this extension. Here is a brief demonstration:
>
> 1. **Technical Feasibility (The "How")**:
>     1. **Generalizing the "State" Definition:Current (Linear)**: In Linear Attention, the "Inter-Chunk State" is a feature map matrix (e.g., $S \in \mathbb{R}^{d \times d}$) that follows a linear recurrence. **Required (Softmax)**: For Softmax attention, the "associative state" that needs to be propagated between chunks consists of normalization statistics (the running maximum $m$ and the running sum $l$) rather than a generative hidden state. Expanding Forge would require allowing the "Inter-Chunk" phase to carry these scalar/vector statistics.
>     2. **Introducing a "Rescaling" Primitive**:Current (Linear): The "Merge" phase in Forge is typically a linear combination at chunk-level. **Required (Softmax)**: The "Merge" phase in Softmax attention requires rescaling. When merging partial results from two blocks, the output of the first block must be scaled down based on the difference between its local maximum and the new global maximum ($O_{new} = O_1 \times e^{m_1 - m_{new}} + \dots$). Adding a rescale(output, old_stats, new_stats) primitive to the Forge DSL would be the key technical step to supporting Softmax.
>
> 2. **The Trade-off: Generality vs. Expressiveness (The "Why Not")**: While technically possible, **we deliberately chose to specialize in Linear Attention to maximize expressiveness**. History suggests that trying to make a DSL "universal" often compromises its ability to handle complex, domain-specific patterns.
>     1. Evidence: A recent work AttentionEngine [1], attempts to support both softmax and linear attention. However, to maintain this generality, its abstraction is too rigid to express advanced linear attention variants like the Delta Rule or Vector-Gated GLA—both of which Forge supports effortlessly.
>     2. Ecosystem: The Softmax domain is already well-served by excellent works like FlexAttention (also, a compiler-driven framework) and FlashAttention-2/3/4. In contrast, Linear Attention researchers are inventing new variants monthly, each requiring a bespoke kernel. Forge solves this specific "N-to-1" compiler challenge, which is currently a larger bottleneck for the community than optimizing Softmax attention. We see greater value in solving the "fragmentation problem" for Linear Attention.
>
> 3. **Transferable Insights**: Finally, while our abstraction is specialized, the system-level innovations we built for Forge are generalizable. Our AOT compilation pipeline and static dispatcher (which eliminate runtime overheads) are universal optimizations that could be directly adopted by Softmax-focused DSLs (like FlexAttention) to improve their performance on short sequences.
>
> [1] Chen, Feiyang, et al. "Attentionengine: A versatile framework for efficient attention mechanisms on diverse hardware platforms." arXiv preprint arXiv:2502.15349 (2025).
>
> ---
>
> Thank you for your candid feedback and for challenging us on the core questions of novelty and scope. Your inquiry regarding the comparison with Softmax attention was particularly insightful; it compelled us to articulate the unique 'fragmentation' problem in Linear Attention and justify our design choices with much stronger theoretical and practical arguments. We believe this discussion has highlighted the distinct value Forge brings to the community. **We hope our responses have alleviated your concerns, and we look forward to any further thoughts you may have!**

---

### Official Review · Reviewer_VUej · 2025-11-03

**Soundness:** 3
**Presentation:** 3
**Contribution:** 3
**Rating:** 4
**Confidence:** 4

**Summary:**

The AI community has been rapidly innovating on AI architectures, but it it painstaking to obtain hardware-efficient implementations. The kernels need to respect the GPU memory hierarchy, support multi-GPU execution, and remain easy to implement. This work observes that there are a few rules underlying linear attentions and encodes these patterns into the Forge DSL.

**Strengths:**

- This is a very important problem space. Many sub-quadratic models have been fast in theory but not in practice, and the challenge of developing kernels has prevented the community from understanding how to obtain the best quality for a fixed wall clock time.
- The paper considers a relevant baseline framework – FLA – and shows consistent speed ups.

**Weaknesses:**

It would be useful to understand how Forge kernels compare to highly optimized kernels, since Triton kernels are routinely slower than CUDA kernels:
- How does Forge generalize to linear attentions that use very large state sizes (e.g., Taylor approximations like ReBased, Based, Learned feature maps with large feature dimension, Mamba-2 with large state size)?  - Does register management, careful use of wgmma/tcgen05, etc., become important?
- How do Forge kernels compare to other DSLs (e.g., TileLang, ThunderKittens which have open-sourced efficient linear attention kernels) in the single-GPU setting? It would be useful to have head-to-head speed comparisons.

The writing would benefit from additional clarity and explanation in certain places:
- Are the kernels for inference/forwards pass or backwards as well? Does Forge consider decoding kernels as well, or prefill-only? It would be useful to clarify and discuss the scope of Forge in the paper, and how the ideas could generalize.
- Do the kernels remain numerically stable for training and inference, for instance as compared to the FLA kernels?
- How do the multi-GPU implementations compare to popular multi-GPU softmax attention implementations? Do we see a speed up from linear attention?
- It would be useful to provide clearer explanation on where the experimental gains arise from: L370-377 attribute it to Forge’s “system level overheads” and “parallelism strategy” but this is a vague explanation

**Questions:**

See above.

---

> ### Author Response · Authors · 2025-11-23
> **Reply part1**
>
> We genuinely appreciate your recognition of Forge as a solution to an important problem space, and we value your detailed feedback and writing suggestions, which helped us improve our manuscript. Below, we address your specific questions.
>
> > ### W1: How Forge handles linear attention with large state sizes ? Does register management, careful use of wgmma/tcgen05, etc., become important?
> 1. **Tiling for Large States**: Forge employs tiling strategy to prevent SRAM overflow based on the fact that different parts of the state within the same time step can be updated in parallel as they have no data dependencies. In the compilation phase, Forge analyzes the state size to generate tiling plan for tensors. Different partitions of the state could either be assigned to separate CTAs for parallel execution, or processed serially within the same CTA (typically bounded by smem capacity) in a pipelined manner. Forge automatically searches for the optimal configuration based on available resources.
> 2. **Decoupled Instruction Management**: Forge does not manually manage registers or smem. Instead, it functions similarly to `torch.compile` (Inductor) by lowering torch code into Triton code. We delegate GPU-specific optimization, such as Tensor Core instruction selection (e.g. wgmma) to the Triton compiler. This decoupled design allows Forge to focus on providing a user-friendly linear-attention specific abstraction and graph-level transformation while leveraging Triton’s mature operator-level compilation for hardware intrinsics. **We added an additional section in Appendix D.2 to discuss hardware sustainability. Thanks for your feedback!**
>
> > ### W2: How do Forge kernels compare to other DSLs?
>
> This is an excellent suggestion that strengthens our evaluation! To address this, we attempted to benchmark both ThunderKittens and TileLang against Forge.
> - ThunderKittens: We made extensive efforts to evaluate tk using its latest main branch. However, we encountered persistent build-time failures (specifically, a missing symbol: `make_causal` error) that prevented us from compiling the linear attention kernels successfully. As a result, we were unable to include it in this specific comparison. **However, we believe tk will offer better performance for experts because it allows for more granular control. Forge, on the other hand, prioritizes user-friendliness. We have added a discussion about tk to the relevant section. (line 186, line 474, line 926)**
> - TileLang: We evaluated TileLang on two representative algorithms: Vanilla Linear Attention and Gated DeltaNet (GDN). To ensure a fair comparison, we utilized the official TileLang examples and aligned the experimental settings strictly with our own (using the same input shapes/dtypes, 25 warmup iterations, and 100 benchmark iterations).
> The head-to-head results on a single H100 GPU are presented below. **We have included related discussions about Tilelang in the main text! (line 186, line 474, line 926)**
>
> Benchmark results on a single H100 GPU are presented below. While Forge achieves competitive or superior latency, we notably observed significant numerical correctness issues (threshold: `rtol=atol=1e-2`) in the current open-source implementation in TileLang repo for Vanilla Linear Attention.
>
> | Algorithm | Seq Length | FLA (ms) | TileLang (ms) | TileLang Mismatch | **Forge (ms)** | **Forge Mismatch** |
> | :--- | :--- | :--- | :--- | :--- | :--- | :--- |
> | **Vanilla LA** | 1024 | 0.31 | 0.067 | 13.40% | **0.157** | **0%** |
> | | 2048 | 0.30 | 0.125 | 14.30% | **0.178** | **0%** |
> | | 4096 | 0.36 | 0.230 | 14.30% | **0.213** | **0%** |
> | | 8192 | 0.44 | 0.448 | 14.60% | **0.319** | **0%** |
> | | 16384 | 0.67 | 0.884 | 14.90% | **0.558** | **0%** |
> | | 32768 | 1.17 | 1.760 | 15.10% | **1.038** | **0%** |
> | **GDN** | 1024 | 0.72 | 0.790 | 0% | **0.140** | **0%** |
> | | 2048 | 0.72 | 0.890 | 0% | **0.220** | **0%** |
> | | 4096 | 0.72 | 0.890 | 0% | **0.420** | **0%** |
> | | 8192 | 0.72 | 0.930 | 0% | **0.720** | **0%** |
> | | 16384 | 1.43 | 1.710 | 0% | **1.430** | **0%** |
> | | 32768 | 2.87 | 3.380 | 0% | **2.850** | **0%** |
>
> *Note: "Mismatch" indicates the relative error and potential correctness issues in current open-source implementation*
>
> Our results highlight a fundamental tradeoff in DSL design: while exposing low-level primitives offers a higher performance ceiling, it often comes with a lower performance floor, making it difficult for non-experts to write efficient or even correct kernels. Conversely, high-level compilers like Forge trade some user control for a significantly higher baseline. By tailoring specifically to the linear attention domain, Forge ensures a high performance floor while retaining maximum flexibility.

---

> ### Author Response · Authors · 2025-11-23
> **Reply Part 2**
>
> > ### Suggestion 1: Clarify the scope. Are the kernels for inference/forwards pass or backwards as well? Does Forge consider decoding kernels?
>
> We apologize for the ambiguity in the initial submission and have updated the manuscript to clarify the scope. **Current Scope (Prefill)** : The current version of Forge focuses on the forward pass during the prefill phase, where parallelization opportunities are highest. **We have update our manualscripts (line 202).** We'd like to discuss how ideas could generalize.
> - Backward Pass: The core abstraction of Forge separates linear attention into three chunk-level computation and we find this abstraction is perfectly suited to allow for the automatic generation of the backward kernel. As demonstrated in recent work[1], implementing several chunk-level computations allows for the derivation of backward passes. We are currently working on the auto-generation of backward kernels based on this principle.
> - Decoding: We currently exclude decoding from Forge's scope. Decoding in linear attention is primarily a one-step state update and memory readout; it is computationally lightweight and offers limited scope for the complex tiling and fusion optimizations that Forge provides for chunk-level parallellism.
>
> [1] [Accelerating Linear Attention Design by Unifying Forward & Backward Propagation](https://openreview.net/forum?id=2gY4aOYR80)
>
> > ### Suggestion 2: List numerical stability of Forge compared to other libraries (e.g. FLA)?
>
> The kernels generated by Forge are numerically stable and achieve parity with FLA.
> We verified this by comparing the output of scalable kernels generated by Forge against FLA kernels (serving as the baseline) across different linear attention variants. We measured the Mean Absolute Value (MAV, $\text{MAV}(O) = \frac{1}{N} \sum_{i=1}^{N} |O_i|$, serving as a baseline for numerical scale) of the outputs and the relative error:  $\text{Error}_{\text{rel}} = \frac{\||O_1-O_2\||_2}{\||O_2\||_2}$ between the two implementations. As shown in the table below, the relative errors are within acceptable tolerance. The input is initialized same as the unittest in FLA.
>
> | Algorithm | FLA Output MAV | Forge Output MAV | Relative Error |
> | :--- | :--- | :--- | :--- |
> | **Vanilla Linear Attention** | `5.3013e-3` | `5.3014e-3` | `2.0e-3` |
> | **Scalar GLA** | `8.1626e-4` | `8.1626e-4` | `2.0e-3` |
> | **DeltaNet (GDN)** | `3.3753e-3` | `3.3753e-3` | `3.0e-3` |
>
> > ### Suggestion 3: Comparison with popular multi-GPU softmax attention implementations.
>
> We added an additional section (**Appendix F**) in our manualscript about comparison with Ring-Attention (implement with FlashAttention-3 and Zig-Zag optimization, which is SOTA of distributed softmax attention operator)[1]. The results show under 8xH100 GPUs, Scalar GLA (e.g. RetNet, Mamaba2) implemented with Forge are orders of magnitude faster than Ring-Attention, especially at longer sequences: e.g., at 512k global sequence length, Forge is over $\mathbf{160\times}$ **faster** than Ring-Attention.
>
> > ### Suggestion 4: It would be useful to provide clearer explanation on where the experimental gains are from.
>
> The performance improvements stem from three specific areas:
> 1. Eliminating Dispatch Overhead: Triton suffers from significant JIT autotuning and Python-side dispatch latency. Forge mitigates this via **Ahead-of-Time (AOT) compilation** and a **static dispatcher**, which eliminates the runtime overhead. This is a primary reason Forge outperforms FLA in short sequences (as shown in ablation study).
> 2. Forge automatically generates code utilizing the **Tensor Memory Accelerator (TMA)** on Hopper GPUs to overlap data transfer with computation. **In contrast, many baseline FLA kernels do not yet leverage TMA instructions**.
> 3. Parallelism Scheduler: Our scheduler dynamically optimizes grid size and parallelism configurations, ensuring higher occupancy than fixed-config kernels. We investigate the auto generated configuration, and find Decoupled Strategy (wont fuse chunk mode and decay mode) performs better only in small shapes: e.g. Scalar GLA with  $\mathrm{B=1, H=1, T=32768}$: `0.418 v.s. 0.459` ms.
> 4. In distributed environments, performance gain from compute-communication fusion, which hides communication latency by overlapping with computation (i.e. state update) at tile granularity while LASP2 incurs significant synchronization and redundancy communication volume.
>
> ---
>
> We are grateful for your detailed review and for highlighting the critical aspects of generalization and scope. Your questions regarding large state management and decoding kernels pushed us to clarify the boundaries and capabilities of Forge. We believe that the additional clarification and experiments on numeric stability and comparison with other DSLs have greatly improved the paper's completeness. **We deeply appreciate your time and positive assessment of our work's potential, and we are happy to provide any further details if needed!**

---

### Author Response · Authors · 2025-11-23
**Rebuttal Summary**

We thank the reviewers for their constructive feedback. We provide a concise summary of our responses to the concerns raised by reviewers **PFK1, r3XC, VUej, and RnXQ**.

## Key Concerns and Our Responses
**1. Scope and Generalization to Softmax Attention (Reviewers `PFK1`, `r3XC`)**
* **Scope**: we have clarified scope of Forge to the prefill phase of linear attention (forward pass) and how can we generalize to bwd pass. (**line 201-204**)
* **Design Choice:** We intentionally specialized Forge for Linear Attention to solve the severe **fragmentation problem** in this domain, prioritizing expressiveness for complex variants (e.g., Delta Rule) over rigid universality.
* **Feasibility:** We clarified that extending Forge to Softmax Attention is theoretically feasible (*Appendix E*). However, we focus on Linear Attention as it lacks the standardized support that softmax attention currently enjoys.

**2. Abstraction Boundaries and Sustainability (Reviewers `VUej`, `RnXQ`, `r3XC`)**
* **Algorithmic Foundational:** 3-phase abstraction in Forge ground in the mathematical property of associativity. It natively supports complex modern variants like **DeltaNet** and **Vector-Gated GLA**. The limitation is non-associative models (e.g., TTT), which fundamentally forego the chunk-wise parallel efficiency Forge targets.
* **Hardware Sustainability:** Forge ensures longevity via **hierarchical decoupling**. By compiling to an intermediate DSL (Triton) rather than raw assembly, Forge automatically inherits upstream support for new hardware and instructions.

**3. Comparison with Related Work (Reviewers `VUej`, `RnXQ`)**
* **TileLang and ThunderKittens:** We added head-to-head benchmarks. Forge achieves comparable or better latency than TileLang on H100 while maintaining **strict numerical correctness** (0% mismatch), whereas the open-source TileLang kernels exhibited >13% relative error in our tests. We add additional discussion about ThunderKittens in revised manuscript.
* **AttentionEngine:** We distinguished Forge by its superior **expressiveness**. While AttentionEngine targets broader generality, its abstraction struggles with data-dependent updates (like the Delta Rule), which Forge handles effortlessly. The comparison with TileLang (backend of AttentionEngine) also shows performance superiority of Forge.
* **Softmax Attention Baseline:** We add an additional experiments comparing with SOTA distributed softmax attention operator in *Appendix F*.
* **Numeric Stability:** We add an additional experiments comparing the numeric stability with FLA. (Reply part1 to `VUej `).

**4. Code Example and Brevity analysis (Reviewers `RnXQ`, `r3XC`)**
* **Code Examples:** We added the code example (Forge frontend and generated code) in *Appendix G* to demonstrate programmability. Using Forge to implement a scalar GLA kernel only involves **50+** lines of code (LoC), which the generated triton code and host launcher is around **1200** LoC.


## Manuscript Update
### Revised Sections
- Additional discussion about related work: ThunderKittens, TileLang (line 186, line 474, line 926). *suggested by Reviewer `VUej`*
- Clarification about speedup in abstract: line 026. *suggested by Reviewer `r3XC`*.
- Scope of Forge (line 201-204). *suggested by Reviewer `VUej`*
### New Sections
- (1) **Why 3-phases abstraction in Forge work?** (2) **How forge achieve sustainability as hardware/infrastructure evolves?**: *Appendix D*.

- (1) **Feasibility of extending Forge to softmax attention.** (2) **Why we choose to specialize in Linear Attention domain?**: *Appendix E*.

- **Additional evaluations**: (1) distributed evaluation on H100 GPU. (2) Comparison with new SOTA distributed softmax attention: *Appendix F*.

- **Code example to demonstrate usage of Forge and the code generated by Forge**: *Appendix G*.
---

We hope this summary assists AC's final assessment!

---

### Meta-Review · Area_Chair_eGe7 · 2026-01-11

**Summary:**

Forge is a domain-specific compiler that generates high-performance kernels for linear attention from a unified 3-phase abstraction (intra-chunk, inter-chunk, merge). This is a timely and impactful contribution addressing the fragmentation problem in linear attention implementations. Strengths include the elegant abstraction grounded in associativity, strong empirical results (1.01-4.9x over FLA, near-linear scaling to 128 GPUs), practical system optimizations (AOT compilation, compute-communication fusion), and reduced engineering burden (50 LoC vs 1200 LoC generated). Initial concerns included: scope limited to forward pass of linear attention only, comparison with other DSLs (TileLang, ThunderKittens), abstraction sustainability for future architectures, and experimental fairness in distributed settings. The rebuttal addressed most concerns with additional experiments, code examples, and clarifications on the principled design choice to specialize for linear attention rather than pursue generality at the cost of expressiveness. If Forge can be extended in the future to include the backward pass (not just forward pass) its impact might be even greater.

**Reviewer Concerns:**

Addressed: (1) Comparison with TileLang showing Forge achieves competitive latency with 0% numerical mismatch vs TileLang's 13%+ error; (2) Scope clarified to prefill/forward pass with discussion of backward pass feasibility; (3) Numerical stability verified against FLA; (4) Added H100 distributed experiments and comparison with Ring-Attention; (5) Code examples added in Appendix G demonstrating 50 LoC input vs 1200 LoC generated; (6) Abstraction sustainability explained via hierarchical decoupling through Triton. Outstanding: backward pass not yet implemented (acknowledged as future work), and limited reviewer engagement post-rebuttal makes it difficult to confirm all concerns fully resolved.

**Reviewer Scores:**

VUej (4): concerns about DSL comparison and scope were addressed, score could potentially increase to 6. PFK1 (4): Did not yet engage in discussion. r3XC (6): Explicitly acknowledged rebuttal addressed concerns, maintained score. RnXQ (6): Did not yet respond post-rebuttal; concerns about abstraction sustainability and code examples were addressed, score likely unchanged or slightly higher.

---

### Decision · Program_Chairs · 2026-01-26

Accept (Poster)